# Meta Stackelberg Game: Robust Federated Learning against Adaptive and Mixed Poisoning Attacks

## Abstract

Recent research has uncovered that federated learning (FL) systems are vulnerable to various security threats. Although various defense mechanisms have been proposed, they are typically non-adaptive and tailored to specific types of attacks, leaving them insufficient in the face of adaptive or mixed attacks. In this work, we formulate adversarial federated learning as a Bayesian Stackelberg Markov game (BSMG) to tackle poisoning attacks of unknown/uncertain types. We further develop an efficient meta-learning approach to solve the game, which provides a robust and adaptive FL defense. Theoretically, we show that our algorithm provably converges to the first-order $\varepsilon$-equilibrium point in $O(\varepsilon^{-2})$ gradient iterations with $O(\varepsilon^{-4})$ samples per iteration. Empirical results show that our meta-Stackelberg framework obtains superb performance against strong model poisoning and backdoor attacks with unknown/uncertain types.

## 1 Introduction

Federated learning (FL) allows multiple devices with private data to jointly train a model without sharing their local data [39]. However, FL systems are vulnerable to various adversarial attacks such as untargeted model poisoning attacks (e.g., IPM [68], LMP [15]) and backdoor attacks (e.g., BFL [2], DBA [71]). To address these vulnerabilities, various robust aggregation rules such as Krum [7], coordinate-wise trimmed mean [69], and FLTrust [10] have been proposed to defend against untargeted attacks, and both training-stage and post-training defenses such as Norm bounding [57], NeuroClip [62], and Prun [64] have been proposed to mitigate backdoor attacks. Further, dynamic defenses that myopically adapt parameters such as learning rate [45], norm clipping threshold [21], and regularizer [1] have been proposed. However, state-of-the-art defenses remain inadequate in countering advanced adaptive attacks (e.g., the reinforcement learning (RL)-based attacks [31, 32]) that dynamically adjust the attack strategy to obtain long-term advantages. Further, current defenses are typically designed to counter specific types of attacks, rendering them ineffective in the presence of mixed attacks. As shown in Table 1 (Section 4), simply combining existing defenses with manual tuning proves ineffective due to the interference between defense methods, the defender's lack of information about adversaries, and the dynamic nature of FL.

In this work, we propose a meta-Stackelberg game (meta-SG) framework that obtains superb defense performance even in the presence of strong adaptive attacks and a mix of attacks of the same or different types (e.g., the coexistence of model poisoning and backdoor attacks). Our meta-SG defense framework is built upon the following key observations. First, when the attack type (to be defined in Section 2.1) is known as priori, the defender can utilize the limited amount of local data at the server and publicly available information to build an approximate world model of the FL system. This allows the defender to identify a robust defense policy offline by solving either a Markov decision process (MDP) when the attack is non-adaptive or a Markov game when the attack is adaptive. This

Submitted to 38th Conference on Neural Information Processing Systems (NeurIPS 2024). Do not distribute.

approach naturally applies to both a single attack and the coexistence of multiple attacks and can potentially produce a (nearly) optimal defense. Second, when the attacks are unknown or uncertain, as in more realistic settings, the problem can be formulated as a Bayesian Stackelberg Markov game (BSMG) [52], which provides a general model for adversarial FL. However, the standard solution concept for BSMG, namely, the Bayesian Stackelberg equilibrium, targets the expected case and does not adapt to the actual attack with an unknown/uncertain type.

Motivated by this limitation, we propose a novel solution concept called meta-Stackelberg equilibrium (meta-SE) for BSMG as a principled way of developing robust and adaptive defenses for FL. By integrating meta-learning and Stackelberg reasoning, meta-SE offers a computationally efficient approach to address information asymmetry in adversarial FL and enables strategic adaptation in online execution in the presence of multiple (adaptive) attackers. Before training an FL model, a meta policy is learned by solving the BSMG using experiences sampled from a set of possible attacks. When facing an actual attacker during FL training, the meta-policy is quickly adapted using a relatively small number of samples collected on the fly. The proposed meta-SG framework only requires a rough estimate of possible worst-case attacks during meta-training, thanks to the generalization ability brought by meta-learning.

To solve the BSMG in the pre-training phase, we propose a meta-Stackelberg learning (meta-SL) algorithm based on the debiased meta-reinforcement learning approach in [14]. The meta-SL provably converges to the first-order $\varepsilon$-approximate meta-SE in $O(\varepsilon^{-2})$ iterations, and the associated sample complexity per iteration is of $O(\varepsilon^{-4})$. Even though meta-SL achieves state-of-the-art sample efficiency presented in [24], its operation involves the Hessian of the defender's value function. To obtain a more practical solution (to bypass the Hessian computation), we further propose a fully first-order pre-training algorithm, called Reptile meta-SL, inspired by Reptile [43]. Reptile meta-SL only utilizes the first-order stochastic gradients from the attacker's and the defender's problem to solve for the approximate equilibrium. The numerical results in Table 1 demonstrate its effectiveness in handling various types of non-adaptive attacks, including mixed attacks , while Figure 2 and Figure 9 highlight its efficiency in coping with uncertain or unknown attacks, including adaptive attacks. Due to the space limit, we move related work section to Appendix A. **Our contributions** are summarized as follows:

- We address critical security problems in FL in the face of attacks that may be adaptive or mixed with multiple types.

- We develop a Bayesian Stackelberg game model (Section 2.2) to capture the information asymmetry in the adversarial FL under multiple uncertain/unknown attacks.

- To create a strategically adaptable defense, we propose a new equilibrium concept: meta-Stackelberg equilibrium (meta-SE), where the defender (the leader) commits to a meta policy and an adaptation strategy, leading to a data-driven approach to tackle information asymmetry.

- To learn the meta equilibrium defense in the pre-training phase, we develop meta-Stackelberg learning (Algorithm 1), an efficient first-order meta RL algorithm, which provably converges to $\varepsilon$-approximate equilibrium in $O(\varepsilon^{-2})$ gradient steps with $O(\varepsilon^{-4})$ samples per iteration, matching the state-of-the-art efficiency in stochastic bilevel optimization.

- We conduct extensive experiments in real-world settings to demonstrate the superb performance of our proposed method.

## 2 Meta Stackelberg Defense Framework

### 2.1 Framework Overview

As shown in Figure 1, the meta-learning framework includes two stages: *pre-training*, *online adaptation*. The *pre-training* stage is implemented in a simulated environment, which allows sufficient training using trajectories generated from the interactions between the defender and the attacker with its type randomly sampled from a set of potential attacks. Both adaptive and non-adaptive attacks could be considered for pre-training. After obtaining a meta-policy, the defender will interact with the real FL environment in the *online adaptation* stage to tune its defense policy using feedback (i.e., model updates and environment parameters) received in the face of real attacks that

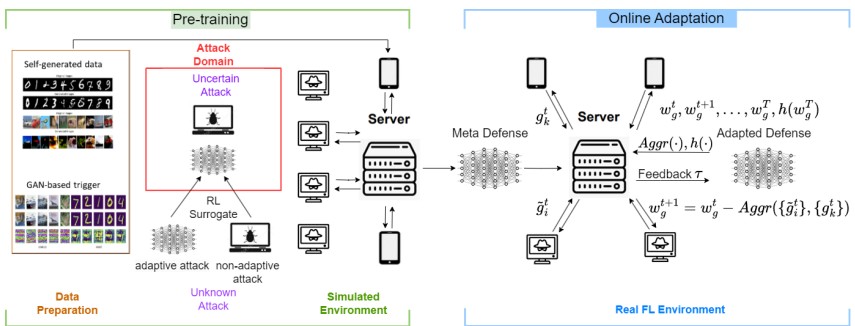

Figure 1: A graphical abstract of meta-Stackelberg defense. In the pertaining stage, a simulated environment is constructed using generated data and the attack domain. The defender utilizes meta-Stackelberg learning (Algorithm 1) to obtain the meta policy to be online adapted in the real FL.

are not necessarily in the pre-training attack set. Finally, at the last round of FL training, the defender will perform a post-training defense on the global model, which may or may not be considered in the design of intelligent attacks. Pre-training and online adaptation are indispensable in the proposed framework. Table 5 in Appendix D indicate that directly applying defense learned from pre-training without online adaptation, as well as adaptation from a randomly initialized defense policy without pre-training, both fail to address malicious attacks.

**FL objective.** Consider a learning system that includes one server and $n$ clients, each client possesses its own private dataset $D_i = (x_i^j, y_i^j)_{j=1}^{|D_i|}$ where $|D_i|$ is the size of the dataset for the $i$-th client. Let $U = \{D_1, D_2, \ldots, D_n\}$ denote the collection of all client datasets. The objective of federated learning is to obtain a model $w$ that minimizes the average loss across all the devices: $\min_w F(w) := \frac{1}{n} \sum_{i=1}^n f(w, D_i)$, where $f(w, D_i) := \frac{1}{|D_i|} \sum_{j=1}^{|D_i|} \ell(w, (x_i^j, y_i^j))$ is the local empirical loss with $\ell(\cdot, \cdot)$ being the loss function.

**Attack objective.** We consider two major categories of attacks: untargeted model poisoning attacks and backdoor attacks. An untargeted model poisoning attack aims to maximize the average model loss, i.e., $\min_w -F(w)$, while a targeted one strives to cause misclassification of poisoned test inputs to one or more target labels (e.g., backdoor attacks). A malicious client $i$ employing targeted attack first produces a poisoned dataset $D_i'$ by altering a subset of data samples $(x_i^j, y_i^j) \in D_i$ to $(\hat{x}_i^j, c^*)$. Here $\hat{x}_i^j$ is the tainted sample with a backdoor trigger inserted, and $c^* \neq y_i^j, c^* \in C$ is the targeted label. Let $\rho_i = |D_i'|/|D_i|$ denote the poisoning ratio, which is typically unknown to the defender. To simplify the notation, we assume that among the $M = M_1 + M_2$ malicious clients, the first $M_1$ malicious clients carry out a targeted attack, and the following $M_2$ malicious clients undertake an untargeted attack. Note that clients in the same category may use different attack methods. Then, the joint objective of these malicious clients is $\min_w F'(w) := \frac{1}{M_1} \sum_{i=1}^{M_1} f(w, D_i') - \frac{1}{M_2} \sum_{i=M_1+1}^{M} f(w, D_i)$.

**FL process.** At each round $t$ out of $H$ rounds of FL training, the server randomly selects a subset of clients $\mathcal{S}^t$ and sends them the most recent global model $w_g^t$. Every benign client $k \in \mathcal{S}^t$ updates the model using their local data via one or more iterations of stochastic gradient descent and returns the model update $g_k^t$ to the server. In contrast, an adversary $j \in \mathcal{S}^t$ creates a malicious model update $\tilde{g}_j^t$ and sends it back. The server then collects the set of model updates $\{\tilde{g}_i^t \cup \tilde{g}_j^t \cup g_k^t\}_{i,j,k \in \mathcal{S}^t}$, for $i \in \{1, \ldots, M_1\}, j \in \{M_1+1, \ldots, M\}, k \in \mathcal{S}^t \setminus [M]$, utilizes an aggregation rule $Aggr$ to combine them, and updates the global model: $w_g^{t+1} = w_g^t - \eta^t Aggr(\tilde{g}_i^t \cup \tilde{g}_j^t \cup g_k^t)$, which is then sent to clients in round $t+1$. At the end of each round, the defender will perform a post-training defense $h(\cdot)$ on the global model $\hat{w}_g^t = h(w_g^t)$ to evaluate the current defense performance. Only at the final round $H$ or whenever a client is leaving the FL systems, the global model with post-training defense $\hat{w}_g^t$ will be sent to all (leaving) clients.

**Attack types.** To simplify the exposition, we assume that a single mastermind attacker controls all malicious clients within the FL system and employs diverse attack strategies on each controlled client. We introduce the concept of *attack type* to differentiate various attack scenarios, which typically include the following three aspects. The first aspect is the attack objective chosen by a malicious client. Let $\Omega_1$ be the set of all possible attack objectives from the defender's knowledge base. We set $\Omega_1 = \{\text{untargeted, targeted}\}$ in this work. The second aspect specifies the attack method (i.e., the

algorithm used to generate the actual attack policy) adopted by a malicious client. Let $\Omega_2$ be the set of all possible attack methods from the defender's knowledge base. The third aspect captures the configuration associated with an attack method, including its hyperparameters and other attributes (e.g., triggers implanted in backdoor attacks, labels used in targeted attacks, and attacker's knowledge about the FL system). Let $\Omega_3$ denote the set of all possible configurations. For each malicious client $i$, the tuple $(\omega_1, \omega_2, \omega_3)_i$ where $\omega_k \in \Omega_k$ for each $k$ fully specifies its particular attack type. Let $\xi = \{(\omega_1, \omega_2, \omega_3)_i\}_{i=1}^{M}$ be the joint attack type. Further, let $\Xi = (\Omega_1 \times \Omega_2 \times \Omega_3)^M$ denote the domain of attacks that the defender is aware of. Table 2 in Appendix C gives the types of all the attacks considered in this work. However, the actual attack type encountered during FL training is not necessary in $\Xi$, although it is presumably similar to a known type in $\Xi$.

## 2.2 Pre-training as a Bayesian Stackelberg Markov game

From the discussion above, the global model updates and the final output are jointly influenced by the defender (through aggregation) and the malicious clients (through corrupted gradients). Hence, the FL process in an adversarial environment can be formulated as a two-player discrete time Bayesian Stackelberg Markov game (BSMG) defined by a tuple $\langle S, A_\mathcal{D}, A_\xi, \mathcal{T}, r, \gamma, H \rangle$. Using discrete time index $t$ (one step corresponds to one FL round), we have the following.

- $S$ is the state space, and its elements represent the global model at each round $s^t = w_g^t$.

- $A_\mathcal{D}$ is the defender's action set. Each action $a_\mathcal{D}^t$ represents a combination of the robust aggregation and post-training defenses: $a_\mathcal{D}^t = \{Aggr(\cdot), h(\cdot)\}$.

- $A_\xi$ is the type-$\xi$ attacker's action set. Each action includes the joint model updates of all malicious clients: $a_\mathcal{A}^t = \{\widetilde{g}_i^t\}_{i=1}^{M_1} \cup \{\widetilde{g}_i^t\}_{i=M_1+1}^{M}$.

- $\mathcal{T}(s^{t+1}|s^t, Aggr(\cdot), a_\mathcal{A}^t)$ specifies the distribution of the next state given the current state and joint actions at $t$, which is determined by the global model update: $w_g^{t+1} = w_g^t - \eta^t Aggr(\widetilde{g}_i^t \cup \widetilde{g}_j^t \cup g_k^t)$.

- $r_\mathcal{D}, r_\xi$ are the defender's and the attacker's reward functions (to be maximized), respectively. The defender aims to minimize the loss after the post-training: $r_\mathcal{D}^t := -F(\widehat{w}_g^t)$ where $\widehat{w}_g^t = h(w_g^t)$. The attacker's $r_\xi^t$ is given by the joint attack objective: $-F'(\widehat{w}_g^t)$.

*Remark* 2.1. The post-training defense is only applied in the final round or to a client leaving the FL system and does not interfere with the model updates on $w_g^t$. The defender's reward function is crafted to encompass post-training, as we prioritize a practical, long-term average reward within an online process, which enables clients to seamlessly join and depart from the FL system. This design enables us to incorporate a post-training defense along with techniques for modifying the model structure, such as drop-off and pruning.

**Simulated environment in the white-box setting.** With the game model defined above, the defender (i.e., the server) can, in principle, identify a strong defense by solving the game (we discuss different solution concepts in Section 3). Due to efficiency and privacy concerns in FL, however, it is often infeasible to solve the game in real time when facing the actual attacker. Instead, the defender can create a simulated environment to approximate the actual FL system during the pre-training stage. The main challenge, however, is that the defender often lacks information about the individual devices in FL. We first consider the *white-box* setting where the defender is aware of the number of malicious devices in each category (i.e., $M_1$ and $M_2$) and their actual attack types, as well as the *non-i.i.d.* level (to be defined in Section 4.1) of local data distributions across devices. However, it does not have access to individual devices' local data and random seeds, making it difficult to simulate clients' local training and evaluate rewards. To this end, we assume that the server has a small amount of root data randomly sampled from the the collection of all client dataset $U$ as in previous work [10, 40]. We then use generative model (e.g., conditional GAN model [41] for MNIST and diffusion model [55] for CIFAR-10 in our experiments) to generate as much data as necessary to mimic the local training (see details in Appendix C). We give an ablation study (Table 6) in Appendix D to evaluate the influence of limited/biased root data. We remark that the purpose of pre-training is to derive a defense policy rather than the model itself. Directly using the shifted data (root or generated) to train the FL model will result in low model accuracy (see Table 5 in Appendix D).

**Handling the black-box setting.** We then consider the more realistic *black-box* setting, where the defender has no access to the number of malicious devices and their actual attack types,

nor the *non-i.i.d.* level of local data distributions. To obtain a robust defense, we assume the server considers the worst-case scenario based on a rough estimate of the missing information (see our ablation study in the experiment section) and adopts the RL-based attacks to simulate the worst-case attacks (see Section 3.1) when the attack is unknown or adaptive. In the face of an unknown backdoor attack, the defender does not know the backdoor triggers and targeted labels. To simulate a backdoor attacker's behavior, we first implement multiple GAN-based attack models as in [12] to generate worst-case triggers (which maximizes attack performance given the backdoor objective) in the simulated environment. Since the defender does not know the poisoning ratio $\rho_i$ and the target label of the attacker's poisoned dataset (needed to determine the attack objective $F'$), we approximate the attacker's reward function by $r_{\mathcal{A}}^t = -F''(\widehat{w}_g^{t+1})$, where

$$F''(w) := \min_{c \in C} \left[ \frac{1}{M_1} \sum_{i=1}^{M_1} \frac{1}{|D_i'|} \sum_{j=1}^{|D_i'|} \ell(w, (\hat{x}_i^j, c)) \right] - \frac{1}{M_2} \sum_{i=M_1+1}^{M} f(\omega, D_i).$$ $F''$ differs $F'$

only in the first $M_1$ clients, where we use a strong target label (that minimizes the expected loss) as a surrogate to the true label $c^*$. We compare the defense performance against white-box and black-box backdoor attacks ( see Figure 10 in Appendix D).

# 3 Meta Stackelberg Learning

Since the pre-training is modeled by a Bayesian Markov Stackelberg game, solving the game efficiently is crucial to a successful defense. This work's main contribution includes the formulation of a new solution concept to the game, meta-Stackelberg equilibrium (meta-SE), and a learning algorithm to approximate such equilibrium in finite time. To motivate the proposed concept, we begin by addressing the defense against non-adaptive attacks.

Consider the attacker employing a non-adaptive attack of type $\xi$; in other words, the attack action at each iteration is determined by a fixed attack strategy $\pi_\xi$, where $\pi_\xi(a)$ gives the probability of taken action $a \in A_\xi$, independent of the FL training and the defense strategy. In this case, BSMG reduces to an MDP, where the transition kernel is $\mathcal{T}_\xi(\cdot|s, a_{\mathcal{D}}) \triangleq \int_{A_\xi} \mathcal{T}(\cdot|s, a_{\mathcal{A}}, a_{\mathcal{D}}) d\pi_\xi(a_{\mathcal{A}})$. Parameterizing the defender's policy $\pi_{\mathcal{D}}(a_{\mathcal{D}}^t|s^t; \theta)$ by a neural network with model weights $\theta \in \Theta$, the solution to the following optimization problem $\max_{\theta \in \Theta} \mathbb{E}_{a_{\mathcal{D}}^t \sim \pi_{\mathcal{D}}, s^t \sim \mathcal{T}_\xi} [\sum_{t=1}^H \gamma^t r_{\mathcal{D}}^t] \triangleq J_{\mathcal{D}}(\theta, \xi)$ gives the optimal defense against the non-adaptive attack. When the actual attack in the online stage falls within $\Xi$, which the defender is uncertain of, one can consider the defense against the expected attack: $\max_\theta \mathbb{E}_{\xi \sim Q} J_{\mathcal{D}}(\theta, \xi)$, where $Q$ is a distribution over the attack domain to be designed by the defender. One intuitive design is to include all reported attack methods in history as the attack domain and their empirical frequency as the $Q$ distribution.

In stark contrast to non-adaptive attacks, an adaptive attack can adjust attack actions to the FL environment and the defense mechanism [31, 32]. Most existing attacks are history-independent [50, 65]. Hence, we assume that an adaptive attack takes the current state (global model) as input, i.e., the attack policy is a Markov policy denoted by $\pi_{\mathcal{A}}(a_{\mathcal{A}}^t|s^t)$. Denoted by $\xi$ the attack type; then, an optimal adaptive attack policy, parameterized by $\phi$, is the best response to the existing defense $\pi_{\mathcal{D}}(\cdot|s^t; \theta)$: $\phi \in \arg\max \mathbb{E}_{a_{\mathcal{A}}^t \sim \pi_\xi, a_{\mathcal{D}}^t \sim \pi_{\mathcal{D}}} [\sum_{t=1}^H \gamma^t r_\xi^t] \triangleq J_{\mathcal{A}}(\theta, \phi, \xi)$. Denote by $\phi_\xi^*$ the maximizer, and then, the defender's cumulative rewards under such attack is $J_{\mathcal{D}}(\theta, \phi_\xi^*, \xi) \triangleq \mathbb{E}_{a_{\mathcal{A}}^t \sim \pi_\xi, a_{\mathcal{D}}^t \sim \pi_{\mathcal{D}}} [\sum_{t=1}^H \gamma^t r_{\mathcal{D}}^t]$.

## 3.1 RL-based attacks and defenses

The actual attack type (which could be either adaptive or non-adaptive) encountered in the online phase may be not in $\Xi$ and thus unknown to the defender. To prepare for these unknown attacks, we propose to use multiple RL-based attacks with different objectives, adapted from RL-based untargeted model poising attack [31] and RL-based backdoor attack [32], as surrogates for unknown attacks, which are added to the attack domain for pre-training. The rationale behind the RL surrogates includes: (1) they achieve strong attack performance by optimizing long-term objectives; (2) they adopt the most general action space (i.e., model updates), which allows them to mimic any adaptive or non-adaptive attacks given the corresponding objectives; (3) they are flexible enough to incorporate multiple attack methods by using RL to tune the hyper-parameters of a mixture of attacks. A similar argument applies to RL-based defenses. We remark that in this paper, an RL-based attack (defense) is not a single attack (defense) as in [31, 32] but a systematically synthesized combination of existing attacks (defenses). In the simulated environment, we train our defense against the strongest white-box

RL attacks in [31, 32] with different objectives (e.g., untargeted or targeted), which is considered the optimal attack strategy. The "worst-case" scenario is commonly used in security scenarios to ensure the associated defense has performance guarantees under "weaker" attacks with similar objectives. Such a robust defense policy gives us a good starting point to further adapt to uncertain or unknown attacks. Our defense is generalizable to other adaptive attacks (see Table 8 in Appendix D). The key novelty of our RL-based defense is that instead of using a fixed and hand-crafted algorithm as in existing approaches, we use RL to optimize the policy network $\pi_{\mathcal{D}}(a_{\mathcal{D}}^t|s^t; \theta)$. Similar to RL-based attacks, the most general action space could be the set of global model parameters. However, the high dimensional action space will lead to an extremely large search space that is prohibitive in terms of training time and memory space. Thus, we apply compression techniques (see Appendix C) to reduce the action from high-dimensional space to a 3-dimensional space. Note that the execution of our defense policy is lightweight, without using any extra data for evaluation/validation. See the discussion in Appendix C on how we apply our RL-based defense during online adaptation.

## 3.2 Meta-Stackelberg equilibrium

As discussed in Section 2.2, one of the key challenges to solving the BSMG is the defender's incomplete information on attack types. Prior works have explored a Bayesian equilibrium approach to address this issue [52]. Given the set of possible attacks $\Xi$ that the defender is aware of and a prior distribution $Q$ over the domain, the Bayesian Stackelberg equilibrium (BSE) is given by the following bi-level optimization:

$$\max_{\theta \in \Theta} \mathbb{E}_{\xi \sim Q}[J_{\mathcal{D}}(\theta, \phi_\xi^*, \xi)] \quad \text{s.t. } \phi_\xi^* \in \arg\max J_{\mathcal{A}}(\theta, \phi, \xi). \tag{BSE}$$

In (BSE), unaware of the exact attacker type, the defender (the leader) aims to maximize the defense performance against an average of all attack types, anticipating their best responses.

From a game-theoretic viewpoint, the Bayesian equilibrium in (BSE) is of ex-ante. The defender determines its equilibrium strategy only knowing the type distribution $Q$. However, as the Markov game proceeds, the attacker's moves (e.g., malicious global model updates) during the interim stage (online stage) reveal additional information on the attacker's private type. This Bayesian equilibrium defense strategy fails to handle the emerging information on the attacker's hidden type in the interim stage, as the policy obtained from (BSE) remains fixed throughout the online stage without adaptation.

To address the limitation of Bayesian equilibrium, we introduce the novel solution concept, meta-Stackelberg equilibrium (meta-SE), to equip the defender with online responsive intelligence under incomplete information. As a synthesis of meta-learning and Stackelberg equilibrium, the meta-SE aims to pre-train a meta policy on a variety of attack types sampled from the attack domain $\Xi$ such that online gradient adaption applied to the base produces a decent defense against the actual attack in the online environment. Using mathematical terms, we denote by $\tau_\xi := (s^k, a_{\mathcal{D}}^k, a_\xi^k)_{k=1}^H$ the trajectory of the FL system under type-$\xi$ attacker up to round $H$, which is subject to the distribution $q(\theta, \xi) := \prod_{t=1}^H \pi_{\mathcal{D}}(a_{\mathcal{D}}^t|s^t; \theta)\pi_\xi(a_{\mathcal{A}}^t|s^t)\mathcal{T}(s^{t+1}|s^t, a_{\mathcal{D}}^t, a_{\mathcal{A}}^t)$. Let $\hat{\nabla}_\theta J_{\mathcal{D}}(\tau)$ be the unbiased estimate of the policy gradient $\nabla_\theta J_{\mathcal{D}}$ using the sample trajectory $\tau_\xi$ (see Appendix E). Then, a one-step gradient adaptation using the sample trajectory is given by $\theta + \eta \nabla_\theta J_{\mathcal{D}}$. Incorporating this gradient adaptation into (BSE) leads to the proposed meta-SE.

$$\max_{\theta \in \Theta} \mathbb{E}_{\xi \sim Q} \mathbb{E}_{\tau \sim q}[J_{\mathcal{D}}(\theta + \eta\hat{\nabla}_\theta J_{\mathcal{D}}(\tau), \phi_\xi^*, \xi)],$$
$$\text{s.t. } \phi_\xi^* \in \arg\max \mathbb{E}_{\tau \sim q} J_{\mathcal{A}}(\theta + \eta\hat{\nabla}_\theta J_{\mathcal{D}}(\tau), \phi, \xi). \tag{meta-SE}$$

The idea of adding the gradient adaptation to the equilibrium is inspired by the recent developments in gradient-based meta-learning [16, 43]. When the attack is non-adaptive, the BSMG reduces to MDP problem, as delineated at the beginning of this section. Consequently, (meta-SE) turns into the standard form of meta-learning [16]. Unlike the conventional (BSE), the solution to (meta-SE) gives the defender a decent defense initialization after pre-training whose gradient adaptation in the online stage is tailored to type $\xi$, since the online trajectory follows the distribution $q(\theta, \xi)$. The novelty of (meta-SE) lies in that the leader (defender) determines an optimal adaptation scheme rather than a policy, which is computed using an online trajectory without knowing the actual type, creating a data-driven strategic adaptation after the pre-training. Besides equation BSE, Appendix G also compares the perfect Bayesian equilibrium with the proposed meta-SE, highlighting the latter's scalability to complex FL systems.

## 3.3 Meta-Stackelberg learning

Unlike finite Stackelberg Markov games that can be solved (approximately) using mixed-integer programming [59] or Q-learning [52], our BSMG admits high-dimensional continuous state and action spaces, posing a more challenging computation issue. Hence, we resort to a two-timescale policy gradient (PG) algorithm, referred to as meta-Stackelberg learning (meta-SL) presented in Algorithm 1, to solve for the meta-SE in a similar vein to [33]. In plain words, meta-SL first learns the attacker's best response at a fast scale (lines 13-15), based on which it updates the defender's meta policy at a slow scale at each iteration using either debiased meta-learning [14] or reptile [43]. The two-timescale meta-SL alleviates the non-stationarity caused by concurrent policy updates from both players [70]. Of particular note is that the debiased meta-learning involves Hessian computation when evaluating the gradient of the defender's objective function since the attacker's best response $\phi_\xi^*(\theta)$ also depends on $\theta$. In contrast, reptile uses a first-order approximation to avoid Hessian. The mathematical subties between two policy gradient estimations are deferred to the Appendix E.

---

**Algorithm 1** Meta-Stackelberg Learning

1: **Input:** the distribution $Q(\Xi)$, initial defense meta policy $\theta^0$, pre-defined attack methods $\{\pi_\xi\}_{\xi \in \Xi}$, pre-trained RL attack policies $\{\phi_\xi^0\}_{\xi \in \Xi}$, step size parameters $\kappa_\mathcal{D}, \kappa_\mathcal{A}, \eta$, and iterations numbers $N_\mathcal{A}, N_\mathcal{D}$;

2: **Output:** $\theta^{N_\mathcal{D}}$;
3: **for** iteration $t = 0$ to $N_\mathcal{D} - 1$ **do**
4:   **if** `meta-RL` (for non-adaptive) **then**
5:     Sample a batch of $K$ attack types $\xi$ from $\Xi$;
6:     Estimate $\hat{\nabla} J_D(\xi) := \hat{\nabla}_\theta J_\mathcal{D}(\theta, \pi_\xi, \xi)|_{\theta = \theta_\xi^t}$;
7:   **end if**
8:   **if** `meta-SG` **then**
9:     Sample a batch of $K$ attack types $\xi \in \Xi$;
10:    **for** each sampled attack $\xi$ **do**
11:      Apply one-step adaptation
      $\theta_\xi^t \leftarrow \theta^t + \eta \hat{\nabla}_\theta J_\mathcal{D}(\theta^t, \phi_\xi^t, \xi)$;
12:      $\phi_\xi^t(0) \leftarrow \phi_\xi^t$;
13:      **for** iteration $k = 0, \ldots, N_\mathcal{A} - 1$ **do**
14:       $\phi_\xi^t(k+1) \leftarrow \phi_\xi^t(k) +$
15:        $\kappa_\mathcal{A} \hat{\nabla}_\phi J_\mathcal{A}(\theta_\xi^t, \phi_\xi^t(k), \xi)$;
16:      **end for**
17:      $\hat{\nabla} J_D(\xi) \leftarrow \hat{\nabla}_\theta J_\mathcal{D}(\theta, \phi_\xi^t(N_\mathcal{A}), \xi)|_{\theta = \theta_\xi^t}$;
18:    **end for**
19:   **end if**
20:   $\theta^{t+1} \leftarrow \theta^t \kappa_\mathcal{D} / K \sum_\xi \hat{\nabla} J_D(\xi)$
21: **end for**

---

The rest of this subsection addresses the computation expense of the proposed meta-SL. We begin with an alternative solution concept for our first-order gradient algorithm, which is slightly weaker than (meta-SE). Let $\mathcal{L}_\mathcal{D}(\theta, \phi, \xi) \triangleq \mathbb{E}_{\tau \sim q} J_\mathcal{D}(\theta + \eta \hat{\nabla}_\theta J_\mathcal{D}(\tau), \phi, \xi), \mathcal{L}_\mathcal{A}(\theta, \phi, \xi) \triangleq \mathbb{E}_{\tau \sim q} J_\mathcal{A}(\theta + \hat{\nabla}_\theta J_\mathcal{D}(\tau), \phi, \xi)$, for a fixed type $\xi \in \Xi$. In the sequel, we will assume $\mathcal{L}_\mathcal{D}$ and $\mathcal{L}_\mathcal{A}$ to be continuously twice differentiable and Lipschitz-smooth with respect to both $\theta$ and $\phi$ as in [33], see Appendix F.

**Definition 3.1.** For $\varepsilon \in (0, 1)$, a pair $(\theta^*, \{\phi_\xi^*\}_{\xi \in \Xi}) \in \Theta \times \Phi^{|\Xi|}$ is a $\varepsilon$-*meta First-Order Stackelbeg Equilibrium* ($\varepsilon$-meta-FOSE) of the meta-SG if it satisfies the following conditions: for $\xi \in \Xi$, $\max_{\theta \in B(\theta^*)} \langle \nabla_\theta \mathcal{L}_\mathcal{D}(\theta^*, \phi_\xi^*, \xi), \theta - \theta^* \rangle \leq \varepsilon$, $\max_{\phi \in B(\phi_\xi^*)} \langle \nabla_\phi \mathcal{L}_\mathcal{A}(\theta^*, \phi_\xi^*, \xi), \phi - \phi_\xi^* \rangle \leq \varepsilon$, where $B(\theta^*) := \{\theta \in \Theta : \|\theta - \theta^*\| \leq 1\}$, and $B(\phi_\xi^*) := \{\phi \in \Phi : \|\phi - \phi_\xi^*\| \leq 1\}$.

Definition 3.1 contains the necessary equilibrium condition for meta-SE in (meta-SE), which can be reduced to $\|\nabla_\theta \mathcal{L}_\mathcal{D}(\theta^*, \phi_\xi, \xi)\| \leq \varepsilon$ and $\|\nabla_\phi \mathcal{L}_\mathcal{A}(\theta^*, \phi_\xi, \xi)\| \leq \varepsilon$ in the unconstraint settings. Since we utilize stochastic gradient in practice, all inequalities mentioned above shall be considered in expectation. The existence of meta-FOSE is guaranteed Theorem F.1 in Appendix F.

Since the value functions $J_\mathcal{A}, J_\mathcal{D}$ are nonconvex, we impose a regularity assumption adapted from the Polyak-Łojasiewicz (PL) condition [26], which is customary in nonconvex analysis. Despite the lack of theoretical justifications for the PL condition in the literature, [33] empirically demonstrates that the cumulative rewards in meta-reinforcement learning satisfy the PL condition, see Figure 4 Appendix D therein. Assumption 3.2 subsequently leads to the main result in Theorem 3.3

**Assumption 3.2** (Stackelberg Polyak-Łojasiewicz condition). There exists a positive constant $\mu$ such that for any $(\theta, \phi) \in \Theta \times \Phi$ and $\xi \in \Xi$, the following inequalities hold: $\frac{1}{2\mu} \|\nabla_\phi \mathcal{L}_\mathcal{D}(\theta, \phi, \xi)\|^2 \geq \max_\phi \mathcal{L}_\mathcal{D}(\theta, \phi, \xi) - \mathcal{L}_\mathcal{D}(\theta, \phi, \xi)$, $\frac{1}{2\mu} \|\nabla_\phi \mathcal{L}_\mathcal{A}(\theta, \phi, \xi)\|^2 \geq \max_\phi \mathcal{L}_\mathcal{A}(\theta, \phi, \xi) - \mathcal{L}_\mathcal{A}(\theta, \phi, \xi)$.

**Theorem 3.3.** *Under assumption 3.2 and other regularity assumptions in Appendix F, for any given $\varepsilon \in (0, 1)$, let the learning rates $\kappa_\mathcal{A}$ and $\kappa_\mathcal{D}$ be properly chosen; let $N_\mathcal{A} \sim \mathcal{O}(\log \epsilon^{-1})$ and $N_b \sim \mathcal{O}(\epsilon^{-4})$ be properly chosen (Appendix F), then, Algorithm 1 finds a $\varepsilon$-meta-FOSE within $N_\mathcal{D} \sim \mathcal{O}(\varepsilon^{-2})$ iterations.*

Finally, we conclude this section by analyzing the meta-SG defense's generalization ability when the learned meta policy is exposed to attacks unseen in the pre-training. Proposition 3.4 asserts that meta-SG is generalizable to the unseen attacks, given that the unseen is not distant from those seen. The formal statement is deferred to Appendix F.

**Proposition 3.4.** *Consider sampled attack types $\xi_1, \ldots, \xi_m$ during the pre-training and the unseen attack type $\xi_{m+1}$ in the online stage. The generalization error is upper-bounded by the "discrepancy" between the unseen and the seen attacks $C(\xi_{m+1}, \{\xi_i\}_{i=1}^m)$.*

# 4 Experiments

## 4.1 Experiment Settings

**Dataset.** Our experiments are conducted on MNIST [30] and CIFAR-10 [28] datasets with a CNN classifier and ResNet-18 model respectively (see Appendix C for details). We consider horizontal FL and adopt the approach introduced in [15] to measure the diversity of local data distributions among clients. Let the dataset encompass $C$ classes, such as $C = 10$ for datasets like MNIST and CIFAR-10. Client devices are divided into $C$ groups (with $M$ attackers evenly distributed among these groups). Each group is allocated $1/C$ of the training samples in the following manner: a training instance labeled as $c$ is assigned to the $c$-th group with a probability of $q \geq 1/C$, while being assigned to every other group with a probability of $(1 - q)/(C - 1)$. Within each group, instances are evenly distributed among clients. A higher value of $q$ signifies a greater *non-i.i.d.* level. By default, we set $q = 0.5$ as the standard *non-i.i.d.* level. We assume the server holds a small amount of root data randomly sampled from the the collection of all client dataset $U$. (100 for MNIST and 200 for CIFAR-10).

**Baseline.** We evaluate our meta-RL and meta-SG defenses under the following untargeted model poisoning attacks including IPM [68] (with scaling factor 2), LMP [15], RL [31], and backdoor attacks including BFL [2] (with poisoning ratio 1), DBA [67] (with 4 sub-triggers evenly distributed to malicious clients and poisoning ratio 0.5), BRL [32], and a mix of attacks from the two categories (see Table 2 for all attacks' categories in Appendix C). We consider various strong defenses as baselines, including training-stage defenses such as Coordinate-wise trimmed mean/median [69], Norm bounding [57], FLTrust [10], Krum [7], and post-training stage defenses such as NeuroClip [62] and Prun [64] and the selected combination of them. We utilize the Twin Delayed DDPG (TD3) [18] algorithm to train both attacker's and defender's policies. We use the following default parameters: number of devices = 100, number of malicious clients for untargeted model poisoning attack = 10, number of malicious clients for backdoor attack = 5 (20 for DBA), client subsampling rate = 10%, number of FL epochs = 500 (1000) for MNIST (CIFAR-10). We fix the initial model and the random seeds for client subsampling and local data sampling for fair comparisons. The details of the experiment setup and additional results are provided in Appendices C and D.

## 4.2 Experiment Results

| Acc/Bac | FedAvg | Trimed Mean | FLTrust | ClipMed | FLTrust+NC | Meta-RL (ours) |
|---------|--------|-------------|---------|---------|------------|----------------|
| NA | 0.7082/0.1 | 0.7093/0.1078 | 0.7139/0.1066 | 0.5280/0.1212 | 0.7100/0.1061 | 0.7053/0.0999 |
| IPM | 0.1369/0.0312 | 0.6542/0.1174 | 0.6828/0.1054 | 0.5172/0.1220 | 0.6656/0.0971 | 0.6862/0.0637 |
| LMP | 0.1115/0.1174 | 0.6224/0.1033 | 0.7071/0.099 | 0.5144/0.121 | 0.7075/0.104 | 0.7109/0.037 |
| BFL | 0.7137/1.0 | 0.7034/1.0 | 0.7145/1.0 | 0.5198/0.5337 | 0.7100/0.1061 | 0.7106/0.0143 |
| DBA | 0.7007/0.7815 | 0.6904/0.7737 | 0.7010/0.8048 | 0.4935/0.6261 | 0.6618/0.9946 | 0.6699/0.2838 |
| IPM+BFL | 0.3104/0.8222 | 0.6415/1.0 | 0.6911/1.0 | 0.5097/0.5776 | 0.6817/0.0267 | 0.6949/0.0025 |
| LMP+DBA | 0.1124/0.1817 | 0.6444/0.7311 | 0.7007/0.7620 | 0.4841/0.6342 | 0.6032/0.8422 | 0.6934/0.2136 |

Table 1: Comparisons of average global model accuracy (acc: higher the better) and backdoor accuracy (bac: lower the better) after 500 rounds under single/multiple type attacks on CIFAR-10. All parameters are set as default and random seeds are fixed.

**Effectiveness against single/multiple type of attacks.** We examine the defense performance of our meta-RL compared with other defense combinations in Table 1 based on average global model accuracy after 500 FL rounds on CIFAR-10, which measures the success of defense and learning speed ignoring the randomness influence (corner-case updates, bias data, etc.) at the bargaining stage of FL. The meta-RL first learns a meta-defense policy from the attack domain involving {NA, IPM,

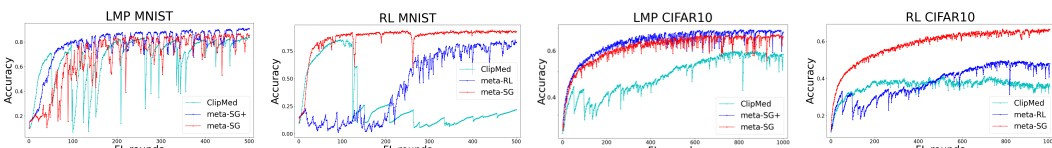

Figure 2: Comparisons of defenses against untargeted model poisoning attacks (i.e., LMP and RL) on MNIST and CIFAR-10. All parameters are set as default and random seeds are fixed.

LMP, BFL, DBA}, then adapts it to the real single/mixed attack. We observe that multiple types of attacks may intervene with each other (e.g., IPM+BFL, LMP+DBA), which makes it impossible to manually address the entangled attacks. It is not surprising to see FedAvg [39] and defenses specifically designed for untargeted attacks (i.e., Trimmed mean, FLTrust) fail to defend backdoor attacks (i.e., BFL, DBA) due to the huge deviation of defense objective from the optimum. For a fair comparison, we further manually tune the norm threshold (more results in Appendix D) from $[0.01, 0.02, 0.05, 0.1, 0.2, 0.5, 1]$ for ClipMed (i.e., Norm bounding + Coordinate-wise Median) and clipping range from $[2 : 2 : 10]$ for FLTrust + NeuroClip to achieve the best performance to balance the global model and backdoor accuracy in linear form (i.e., Acc - Bac). Intuitively, a tight threshold/range has better performance in defending against backdoor attacks, yet will hinder or even damage the FL progress. On the other hand, a loose threshold/range fails to defend backdoor injection. Nevertheless, manually tuning in real-world FL scenarios is nearly impossible due to the limited knowledge of the ongoing environment and the presence of asymmetric adversarial information. Instead of suffering from the above concerns and exponential growth of parameter combination possibilities, our data-driven meta-RL approach can automatically tune multiple parameters at each round. Targeting the cumulative defense rewards, the RL approach naturally holds more flexibility than myopic optimization.

**Adaptation to uncertain/unknown attacks.** To evaluate the necessity and efficiency of adaptation from the meta-SG policy in the face of unknown attacks, we plot the global model accuracy graph over FL epochs. The meta-RL pre-trained from non-adaptive attack domain {NA, IPM, LMP, BFL, DBA} (RL attack is unknown), while meta-SG pre-train from interacting with a group of RL attacks initially target on {FedAvg, Coordinate-wise Median, Norm bounding, Krum, FLTrust } (LMP is unknown). The meta-SG plus (i.e., meta-SG+) is a pre-trained model from the combined attack domain of the above two. All three defenses then adapt to the real FL environments under LMP or RL attacks. As shown in Figure 2, the meta-SG can quickly adapt to both uncertain RL-based adaptive attack (attack action is time-varying during FL) and unknown LMP attack, while meta-RL can only slowly adapt to or fail to adapt to the unseen RL-based adaptive attacks on MNIST and CIFAT-10 respectively. In addition, the first and the third Figures in Figure 2 demonstrate the power of meta-SG against unknown LMP attacks, even if LMP is not directly used during its pre-training stage. The results are only slightly worse than meta-SG plus, where LMP is seen during pre-training. Similar observations are given under IPM in Appendix D.

# 5 Conclusion

We have proposed a meta-Stackelberg framework to tackle attacks of uncertain or unknown types in federated learning through data-driven adaptation. The proposed meta-Stackelberg learning approach is computationally tractable and strategically adaptable, targeting mixed and adaptive attacks under incomplete information. The major limitation of our current approach pertains to privacy concerns. Our current simulation necessitates that the defender either accesses a small portion of root data or learns clients' data through inversion, which slightly violates the privacy principles of FL. To minimize privacy risks, we train our meta-policy in a simulated environment and apply data augmentation to blur the learned data. In our experiments, the current "black-box" setting operates under certain conditions: we test only one or a few agnostic variables at a time while leaving other information known to the defender (see Appendix D). In our future work, we plan to incorporate additional state-of-the-art defense algorithms to counter more potent attacks, such as edge-case attacks [63], as well as other attack types, such as privacy-leakage attacks [37]. We will also explore new application scenarios, including NLP and large generative models. Our framework could be further improved by including a client-side defense mechanism that closely mirrors real-world scenarios, replacing the current processes of self-data generation.

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

## A Related Works

**Poisoning/backdoor attacks and defenses in FL** Several defensive strategies against model poisoning attacks broadly fall into two categories. The first category includes robust-aggregation-based defenses encompassing techniques such as dimension-wise filtering. These methods treat each dimension of local updates individually, as explored in studies by [4, 69]. Another strategy is client-wise filtering, aiming to limit or entirely eliminate the influence of clients who might harbor malicious intent. This approach has been examined in the works of [7, 47, 57]. Some defensive methods necessitate the server having access to a minimal amount of root data, as detailed in the study by [10]. Naive backdoor attacks are limited by even simple defenses like norm-bounding [57] and weak differential private [20] defenses. Despite the sophisticated design of state-of-the-art non-addaptive backdoor attacks against federated learning, post-training stage defenses [64, 42, 49] can still effectively erase suspicious neurons/parameters in the backdoored model.

**Incomplete Information in Adversarial Machine Learning** Prior works have attempted to tackle the challenge of incomplete information through two distinct approaches. The first approach is the "infer-then-counter" approach, where the hidden information regarding the attacks is first inferred through observations. For example, one can infer the backdoor triggers through reverse engineering using model weights [60], based on which the backdoor attacks can be mitigated [72]. The inference helps adapt the defense to the present malicious attacks. However, this inference-based adaptation requires prior knowledge of the potential attacks (i.e., backdoor attacks) and does not directly lend itself to mixed/adaptive attacks. Moreover, the inference and adaptation are offline, unable to counter online adaptive backdoor attack [31]. The other approach explored the notion of robustness that prepares the defender for the worst case [54, 52], which often leads to a Stackelberg game (SG) between the defender and the attacker. Yet, such a Stackelberg approach often leads to conservative defense, lacking adaptability.

## B Broader Impact

**Towards Universal Robust Federated Learning.** Our goal is to establish a comprehensive framework for universal federated learning defense against all kinds of attacks. This framework ensures that the server remains oblivious to any details pertaining to the environment or potential attackers. Still, it possesses the ability to swiftly adapt and respond to uncertain or unknown attackers during the actual federated learning process. Nevertheless, achieving this universal defense necessitates an extensive attack set through pre-training, which often results in a protracted convergence time toward a meta-policy. Moreover, the effectiveness and efficiency of generalizing from a wide range of diverse distributions pose additional challenges. Considering these, we confine our experiments in this paper to specifically address a subset of uncertainties and unknowns. This includes variables such as the method of attacker, the number of attackers, the level of independence and identically distributed data, backdoor triggers, backdoor targets, and other relevant aspects. However, we acknowledge that our focus is not all-encompassing, and there may be other factors that remain unexplored in our research.

**Meta Equilibrium and Information Asymmetry.** Information asymmetry is a prevailing phenomenon arising in a variety of contexts, including adversarial machine learning (e.g. FL discussed in this work), cyber security [38], and large-scale network systems [34]. Our proposed meta-equilibrium offers a data-driven approach tackling asymmetric information structure in dynamic games without Bayesian-posterior beliefs. Achieving the strategic adaptation through stochastic gradient descent, the meta-equilibrium is computationally superior to perfect Bayesian equilibrium and better suited

for real-world engineering systems involving high-dimensional continuous parameter spaces. It is expected that the meta-equilibrium can also be relevant to other adversarial learning contexts, cyber defense, and decentralized network systems.

# C  Experiment Setup

**Datasets.**   We consider two datasets: MNIST [30] and CIFAR-10 [28], and default $i.i.d.$ local data distributions, where we randomly split each dataset into $n$ groups, each with the same number of training samples. MNIST includes 60,000 training examples and 10,000 testing examples, where each example is a 28×28 grayscale image, associated with a label from 10 classes. CIFAR-10 consists of 60,000 color images in 10 classes of which there are 50,000 training examples and 10,000 testing examples. For the *non-i.i.d.* setting (see Figure 11(d)), we follow the method of [15] to quantify the heterogeneity of the data. We split the workers into $C = 10$ (for both MNIST and CIFAR-10) groups and model the *non-i.i.d.* federated learning by assigning a training instance with label $c$ to the $c$-th group with probability $q$ and to all the groups with probability $1 - q$. A higher $q$ indicates a higher level of heterogeneity.

**Federated Learning Setting.**   We use the following default parameters for the FL environment: local minibatch size = 128, local iteration number = 1, learning rate = 0.05, number of workers = 100, number of backdoor attackers = 5, number of untargeted model poisoning attackers = 20, subsampling rate = $10\%$, and the number of FL training rounds = 500 (resp. 1000) for MNIST (resp. CIFAR-10). For MNIST, we train a neural network classifier of 8×8, 6×6, and 5×5 convolutional filter layers with ReLU activations followed by a fully connected layer and softmax output. For CIFAR-10, we use the ResNet-18 model [22]. We implement the FL model with PyTorch [46] and run all the experiments on the same 2.30GHz Linux machine with 16GB NVIDIA Tesla P100 GPU. We use the cross-entropy loss as the default loss function and stochastic gradient descent (SGD) as the default optimizer. For all the experiments except Figures 11(c) and 11(d), we fix the initial model and random seeds of subsampling for fair comparisons.

**Baselines.**   We evaluate our defense method against various state-of-the-art attacks, including non-adaptive and adaptive untargeted model poison attacks (i.e., IPM [68], LMP [15], RL [31]), as well as backdoor attacks (BFL [2] without model replacement, BRL [32], with tradeoff parameter $\lambda = 0.5$, DBA [67] where each selected attacker randomly chooses a sub-trigger as shown in Figures 6, PGD attack [63] with a projection norm of 0.05), and a combination of both types. To establish the effectiveness of our defense, we compare it with several strong defense techniques. These baselines include defenses implemented during the training stage, such as Krum [7], ClipMed [69, 57, 31] (with norm bound 1), FLTrust [10] with 100 root data samples and bias $q = 0.5$, training stage CRFL [66] with norm bound of 0.02 and noise level $1e - 3$ as well as post-training defenses like NeuroClip [62] and Prun [64]. We use the original clipping thresholds 7 in [62] and set the default Prun number to 256.

| Attack type | Category | Adaptivity |
|---|---|---|
| IPM [68] | untargeted model poisoning | non-adaptive |
| LMP [15] | untargeted model poisoning | non-adaptive |
| BFL [2] | backdoor | non-adaptive |
| DBA [67] | backdoor | non-adaptive |
| RL [31] | untargeted model poisoning | adaptive |
| BRL [32] | backdoor | adaptive |

Table 2: A table showcasing all attacks in the experiments, with their corresponding categories and adaptivities.

**Reinforcement Learning Setting.**   In our RL-based defense, since both the action space and state space are continuous, we choose the state-of-the-art Twin Delayed DDPG (TD3) [18] algorithm to individually train the untargeted defense policy and the backdoor defense policy. We implement our simulated environment with OpenAI Gym [9] and adopt OpenAI Stable Baseline3 [48] to implement TD3. The RL training parameters are described as follows: the number of FL rounds = 300 rounds, policy learning rate = 0.001, the policy model is MultiInput Policy, batch size = 256, and $\gamma = 0.99$ for updating the target networks. The default $\lambda = 0.5$ when calculating the backdoor rewards.

| Settings | Pre-training | Online-adaptation | Related figures/tables |
|---|---|---|---|
| meta-RL | {NA, IPM, LMP, BFL, DBA} | {IPM, LMP, BFL, DBA, IPM+BFL, LMP+DBA} | Table 1,Figures 2, 9 and 11 |
| meta-SG | {RL, BRL} | {IPM, LMP, RL, BRL} | Tables 4 and 8,Figures 2 and 9 to 11 |
| meta-SG+ | {NA, IPM, LMP, BFL, DBA, RL, BRL} | {IPM, LMP, RL, BRL} | Figures 2 and 9 |

Table 3: A table showcasing the attacks and defenses employed during pre-training and online-adaptation, with links to the relevant figures or tables. RL and BRL are initially target on {FedAvg, ClipMed, Krum, FLTrust+NC} during pre-training.

**Meta-learning Setting.** The attack domains (i.e., potential attack sets) are built as following: For meta-RL, we consider IPM [68], LMP [15], EB [5] as three possible attack types. For meta-SG against untargeted model poisoning attack, we consider RL-based attacks [31] trained against Krum [7] and ClipMed [31, 69, 57] as initial attacks. For meta-SG against backdoor attack, we consider RL-based backdoor attacks [32] trained against Norm-bounding [57] and NeuroClip [62] (Prun [64]) as initial attacks. For meta-SG against mix type of attacks, we consider both RL-based attacks [31] and RL-based backdoor attacks [32] described above as initial attacks.

At the pre-training stage, we set the number of iterations $T = 100$. In each iteration, we uniformly sample $K = 10$ attacks from the attack type domain (see Algorithm 2 and Algorithm 1). For each attack, we generate a trajectory of length $H = 200$ for MNIST ($H = 500$ for CIFAR-10), and update both attacker's and defender's policies for 10 steps using TD3 (i.e., $l = N_\mathcal{A} = N_\mathcal{D} = 10$). At the online adaptation stage, the meta-policy is adapted for 100 steps using TD3 with $T = 10$, $H = 100$ for MNIST ($H = 200$ for CIFAR-10), and $l = 10$. Other parameters are described as follows: single task step size $\kappa = \kappa_\mathcal{A} = \kappa_\mathcal{D} = 0.001$, meta-optimization step size $= 1$, adaptation step size $= 0.01$.

**Space Compression.** Following the BSMG model, it is most generally to use $w_g^t$ or $(w_g^t, \mathbf{I}^t)$ as the state, and $\{\tilde{g}_k^t\}_{k=1}^{M_1+M_2}$ or $w_g^{t+1}$ as the action for the attacker and the defender, respectively, if the federated learning model is small. However, when we use federated learning to train a high-dimensional model (i.e., a large neural network), the original state/action space will lead to an extremely large search space that is prohibitive in terms of training time and memory space. We adopt the RL-based attack in [31] to simulate an adaptive model poisoning attack and the RL-based local search in [32] to simulate an adaptive backdoor attack, both having a 3-dimensioanl real action spaces after space comparison (see ). We further restrict all malicious devices controlled by the same attacker to take the same action. To compress the state space, we reduce $w_g^t$ to only include its last two hidden layers for both attacker and defender and reduce $\mathbf{I}^t$ to the number of malicious clients sampled at round $t$.

Our approach rests on an RL-based synthesis of existing specialized defense methods against mixed attacks, where multiple defenses can be selected at the same time and combined with dynamically tuned hyperparameters. The following specialized defenses are selected in our implementation. For training stage aggregation-based defenses, we first normalize the magnitude of all gradients to a threshold $\alpha \in (0, \max_{i \in \mathcal{S}^t} \{\|g_i^t\|\})$, then apply coordinate-wise trimmed mean [69] with trimmed rate $\beta \in [0, 1)$. For post-training defense, NeuroClip [62] with clip range $\varepsilon$ or Prun [64] with mask rate $\sigma$ is applied. The concrete approach used in each of the above defenses can be replaced by other defense methods. The key novelty of our approach is that instead of using a fixed and hand-crafted algorithm as in existing approaches, we use RL to optimize the policy network $\pi_\mathcal{D}(a_\mathcal{D}^t|s^t; \theta)$. Similar to RL-based attacks, the most general action space could be the set of global model parameters. However, the high dimensional action space will lead to an extremely large search space that is prohibitive in terms of training time and memory space. Thus, we apply reduce the action space to $a_\mathcal{D}^t := (\alpha^t, \beta^t, \varepsilon^t/\sigma^t)$. Note that the execution of our defense policy is lightweight, without using any extra data for evaluation/validation.

**Self-generated Data.** We begin by acknowledging that the server only holds a small amount of initial data (200 samples with $q = 0.1$ in this work) learned from first 20 FL rounds using inverting gradient [19], to simulate training set with 60,000 images (for both MNIST and CIFAR-10) for FL. This limited data is augmented using several techniques such as normalization, random rotation, and color jittering to create a larger and more varied dataset, which will be used as an input for generative models.

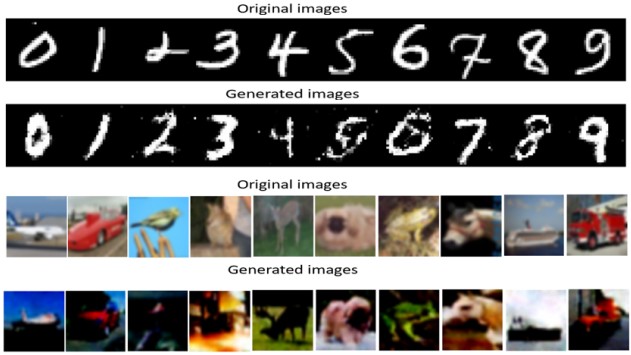

Figure 3: Self-generated MNIST images using conditional GAN [41] (second row) and CIFAR-10 images using a diffusion model [55] (fourth row).

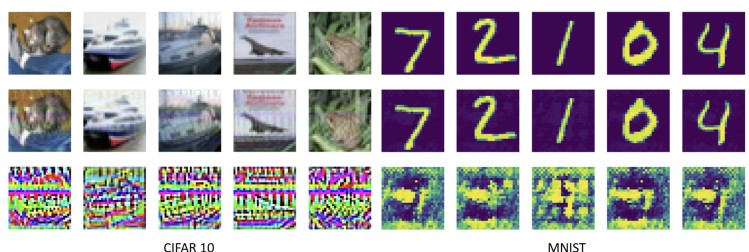

Figure 4: Generated backdoor triggers using GAN-based models [12]. Original image (first row). Backdoor image (second row). Residual (third row).

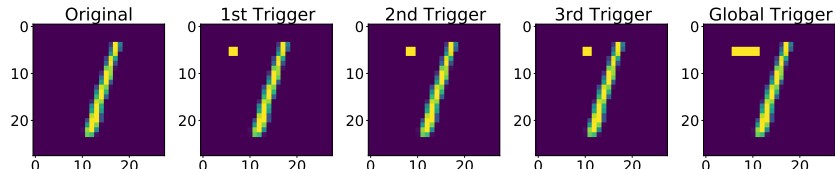

Figure 5: MNIST backdoor trigger patterns. The global trigger is considered the default poison pattern and is used for backdoor accuracy evaluation. The sub-triggers are used by pre-training and DBA only.

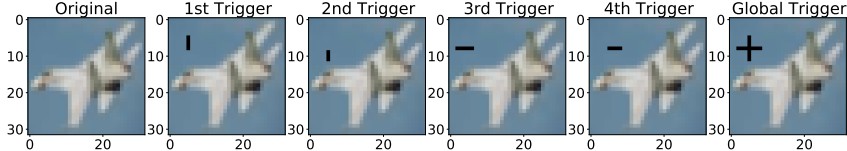

Figure 6: CIFAR-10 fixed backdoor trigger patterns. The global trigger is considered the default poison pattern and is used for online adaptation stage backdoor accuracy evaluation. The sub-triggers are used by pre-training and DBA only.

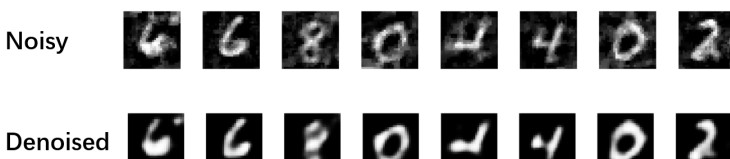

Noisy

Denoised

Figure 7: Examples of reconstructed images using inverting gradient (before and after denoising)

For MNIST, we use the augmented dataset to train a Conditional Generative Adversarial Network (cGAN) model [41, 44] built upon the codebase in [29]. The cGAN model for the MNIST dataset comprises two main components - a generator and a discriminator, both of which are neural networks. Specifically, we use a dataset with 5,000 augmented data as the input to train cGAN, keep the network parameters as default, and set the training epoch as 100.

For CIFAR-10, we leverage a diffusion model implemented in [11] that integrates several recent techniques, including a Denoising Diffusion Probabilistic Model (DDPM) [23], DDIM-style deterministic sampling [56], continuous timesteps parameterized by the log SNR at each timestep [27] to enable different noise schedules during sampling. The model also employs the 'v' objective, derived from Progressive Distillation for Fast Sampling of Diffusion Models [51], enhancing the conditioning of denoised images at high noise levels. During the training process, we use a dataset with 50,000 augmented data samples as the input to train this model, keep the parameters as default, and set the training epoch as 30.

**Simulated Environment.** To further improve efficiency and privacy, the defender simulate a smaller FL system when solving the game. In our experiments, we include 10 clients in pre-training while using 100 clients in the online FL system. The simulation relies on a smaller dataset (generated from root data) and endures a shorter training time (100 (500) FL rounds for MINST (CIFAR-10) v.s. 1000 rounds in online FL experiments). Although the offline simulated Markov game deviates from the ground truth, the learned meta-defense policy can quickly adapt to the real FL during the online adaptation, as shown in our experiment section.

**Backdoor Attacks.** We consider the trigger patterns shown in Figure 4 and Figure 6 for backdoor attacks. For triggers generated using GAN (see Figure 4), the goal is to classify all images of different classes to the same target class (all-to-one). For fixed patterns (see Figure 6), the goal is to classify images of the airplane class to the truck class (one-to-one). The default poisoning ratio is 0.5 in both cases. The global trigger in Figure 6 is considered the default poison pattern and is used for the online adaptation stage for backdoor accuracy evaluation. In practice, the defender (i.e., the server) does not know the backdoor triggers and targeted labels. To simulate a backdoor attacker's behavior, we first implement multiple GAN-based attack models as in [12] to generate worst-case triggers (which maximizes attack performance given backdoor objective) in the simulated environment. Since the defender does not know the poisoning ratio $\rho_i$ and target label of the attacker's poisoned dataset (involved in the attack objective $F'$), we approximate the attacker's reward function as $r_{\mathcal{A}}^t = -F''(\widehat{w}_g^{t+1})$, $F''(w) := \min_{c \in C} \left[ \frac{1}{M_1} \sum_{i=1}^{M_1} \frac{1}{|D'_i|} \sum_{j=1}^{|D'_i|} \ell(w, (\hat{x}_i^j, c)) \right] - \frac{1}{M_2} \sum_{i=M_1+1}^{M} f(\omega, D_i)$. $F''$ differs $F'$ only in the first $M_1$ clients, where we use a strong target label (the minimizer) as a surrogate to the true label $c^*$.

**Inverting Gradient/Reverse Engineering.** In invert gradient, we set the step size for inverting gradients $\eta' = 0.05$, the total variation parameter $\beta = 0.02$, optimizer as Adam, the number of iterations for inverting gradients $max\_iter = 10,000$, and learn the data distribution from scratch. The number of steps for distribution learning is set to $\tau_E = 100$. 32 images are reconstructed (i.e., $B' = 32$) and denoised in each FL epoch. If no attacker is selected in the current epoch, the aggregate gradient estimated from previous model updates is reused for reconstructing data. To build the denoising autoencoder, a Gaussian noise sampled from $0.3\mathcal{N}(0,1)$ is added to each dimension of images in $D_{reconstructed}$, which are then clipped to the range of [0,1] in each dimension. The result is shown in Figure 7.

In the process of reverse engineering, we use Neural Cleanse [61] to find hidden triggers (See Figure 8) connected to backdoor attacks. This method is essential for uncovering hidden triggers

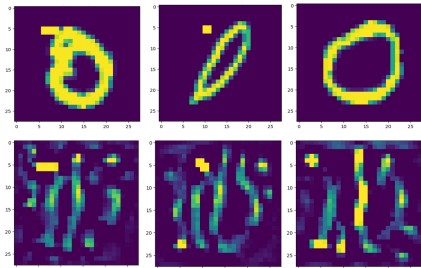

Figure 8: Reversed MNIST backdoor trigger patterns. Original triggers (first row). Reversed triggers (second row)

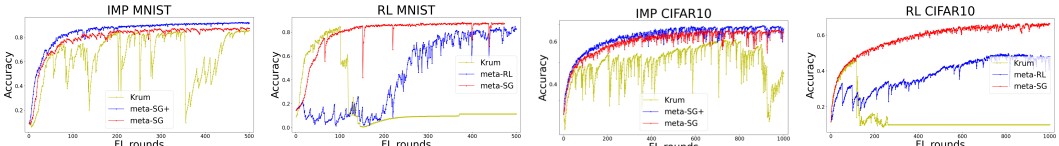

Figure 9: Comparisons of defenses against untargeted model poisoning attacks (i.e., IPM and RL) on MNIST and CIFAR-10. RL-based attacks are trained before FL round 0 against the associate defenses (i.e., Krum and meta-policy of meta-RL/meta-SG). All parameters are set as default and all random seeds are fixed.

and for preventing such attacks. In particular, we use the global model, root generated data and inverted data as inputs to reverse backdoor triggers. The Neural Cleanse class from ART is used for this purpose. The reverse engineering process in this context involves using the generated backdoor method from the Neural Cleanse defense to find the trigger pattern that the model is sensitive to. The returned pattern and mask can be visualized to understand the nature of the backdoor.

**Online Adaptation and Execution.** During the online adaptation stage, the defender starts by using the meta-policy learned from the pre-training stage to interact with the true FL environment, while collecting new samples $\{s, a, \widetilde{r}, s'\}$. Here, the estimated reward $\widetilde{r}$ is calculated using the self-generated data and simulated triggers from the pertaining stage, as well as new data inferred online through methods such as inverting gradient [19] and reverse engineering [61]. Inferred data samples are blurred using data augmentation [53] while protecting clients' privacy. For a fixed number of FL rounds (e.g., 50 for MNIST and 100 for CIFAR-10 in our experiments), the defense policy will be updated using gradient ascents from the collected trajectories. Ideally, the defender's adaptation time (including the time for collecting new samples and that for updating the policy) should be significantly less than the whole FL training period so that the defense execution will not be delayed. In real-world FL training, the server typically waits for up to 10 minutes before receiving responses from the clients [8, 25], enabling defense policy's online update with enough episodes.

## D Additional Experiment Results

**More untargetd model poisoning/backdoor results.** As shown in Figure 9, similar to results in Figure 2 as described in Section 4, meta-SG plus achieves the best performance (slightly better than meta-SG) under IPM attacks for both MNIST and CIFAR-10. On the other hand, meta-SG performs the best (significantly better than meta-RL) against RL-based attacks for both MNIST and CIFAR-10. Notably, Krum can be easily compromised by RL-based attacks by a large margin. In contrast, meta-RL gradually adapts to adaptive attacks, while meta-SG displays near-immunity against RL-based attacks. In addition, we illustrate results under backdoor attacks and defenses on MNIST in Table 4.

**Defender's knowledge of backdoor attacks.** We consider two settings: 1) the server knows the backdoor trigger but is uncertain about the target label, and 2) the server knows the target label but not the backdoor trigger. In the former case, the meta-SG first pre-trains the defense policy with RL attacks using a known fixed global pattern (see Figure 6) targeting all 10 classes in CIFAR-10, then adapts with an RL-based backdoor attack using the same trigger targeting class 0 (airplane), with

| Bac | Krum | CRFL | Meta-SG (ours) |
|-----|------|------|----------------|
| BFL | 0.8257 | 0.4253 | 0.0086 |
| DBA | 0.4392 | 0.215 | 0.2256 |
| BRL | 0.9901 | 0.8994 | 0.2102 |

Table 4: Comparisons of average backdoor accuracy (lower the better) after 250 FL rounds under backdoor attacks and defenses on MNIST. All parameters are set as default and all random seeds are fixed.

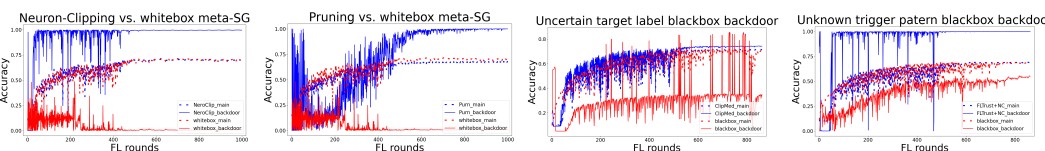

Figure 10: Comparisons of baseline defenses, i.e., NeuroClip, Prun, ClipMed, FLTrust+NeuroClip (from left to right) and whitebox/blackbox meta-SG under RL-based backdoor attack (BRL) on CIFAR-10. The BRLs are trained before FL round 0 against the associate defenses (i.e., NeuroClip, Prun, ClipMed, FLTrust+NC and meta-policy of meta-SG). Other parameters are set as default and all random seeds are fixed.

results shown in the third figure of Figure 10. In the latter case where the defender does not know the true backdoor trigger used by the attacker, we implement the GAN-based model [12] to generate the worst-case triggers (see Figure 4) targeting one known label (truck). The meta-SG will train a defense policy with the RL-based backdoor attacks using the worst-case triggers targeting the known label, then adapt with a RL-based backdoor attack using a fixed global pattern (see Figure 6) targeting the known label in the real FL environment (results shown in the fourth graph in Figure 10. We call the two above cases **blackbox** settings since the defender misses key backdoor information and solely depends on their own generated data/triggers w/o inverting/reversing during online adaptation. In the **whitebox** setting, the server knows the backdoor trigger pattern (global) and the targeted label (truck), and is trained by true clients' data. The corresponding results are in the first two graphs of Figures 10, which show the upper bound performance of meta-SG and may not be practical in a real FL environment. Post-training defenses alone (i.e., NeuroClip and Prun) and combined defenses (i.e., ClipMed and FLTrust+NC) are susceptible to RL-based attacks once the defense mechanism is known. On the other hand, as depicted in Figure 10, we demonstrate that our whitebox meta-SG approach is capable of effectively eliminating the backdoor influence while preserving high main task accuracy simultaneously, while blackbox meta-SG against uncertain labels is unstable since the meta-policy will occasionally target a wrong label, even with adaptation and blackbox meta-SG against unknown trigger is not robust enough as its backdoor accuracy still reaches nearly $50\%$ at the end of FL training.

| Acc | NA/FedAvg | Root data | Generated data | Pre-train only | Online-adapt only |
|-----|-----------|-----------|----------------|----------------|-------------------|
| MNIST | 0.9016 | 0.4125 | 0.5676 | 0.6125 | 0.4134 |
| CIFAR-10 | 0.7082 | 0.2595 | 0.3833 | 0.1280 | 0.3755 |

Table 5: Ablation studies of only using root data/generated dataset in simulated environment to learn the FL model and the defense performance under IPM of directly applying meta-policy learned from pre-training without adaptation/starting online adaptation from a randomly initialized defense policy. Results are average globel model accuracy after 250 (500) FL rounds on MNIST (CIFAR-10). All parameters are set as default and all random seeds are fixed..

**Importance of inverting/reversing methods.** In the ablation study, we examine a practical and relatively well-performed **graybox** meta-SG. The graybox meta-SG has the same setting as **blackbox** meta-SG during pre-training as describe in Section 2.2, but utilizes inverting gradient [19] and reverse engineering [61] during online adaptation to learn clients' data and backdoor trigger in a way without breaking the privacy condition in FL. The graybox approach only learns ambiguous data from clients, then applies data augmentation (e.g., noise, distortion) and combines them with previously generated data before using. Figure 11(a) illustrates that graybox meta-SG exhibits a more stable and robust mitigation of the backdoor attack compared to blackbox meta-SG. Furthermore, in Figure 11(b),

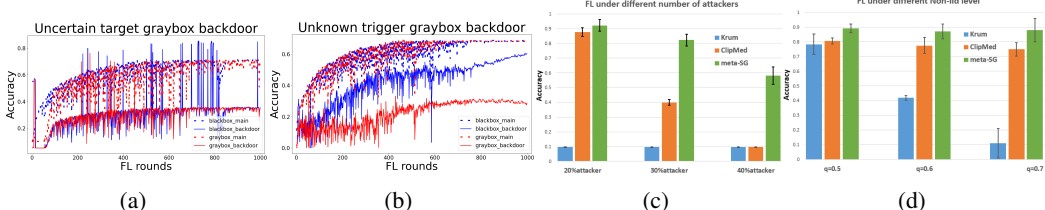

|  | (a) | (b) | (c) | (d) |

Figure 11: Ablation studies. (a)-(b): uncertain backdoor target and unknown backdoor triggers, where the meta-policies are trained by worst-case triggers generated from GAN-based models [12] or targeting multiple labels on CIFAR-10 during pre-training and utilizing inverting gradient [19] and reverse engineering [61] during online adaptation. (c)-(d): meta-RL tested by the number of malicious clients in [20%, 30%, 40%] and non-$i.i.d.$ level in $q = [0.5, 0.6, 0.7]$ on MNIST compared with Krum and ClipMed under LMP attack. Other parameters are set as default.

graybox meta-SG demonstrates a significant reduction in the impact of the backdoor attack, achieving nearly a 70% mitigation, outperforming blackbox meta-SG.

**Number of malicious clients/Non-i.i.d. level.** Here we apply our meta-RL to study the impact of inaccurate knowledge of the number of malicious clients and the non-$i.i.d.$ level of clients' local data distribution. With rough knowledge that the number of malicious clients is in the range of 5%-50%, the meta-SG will pre-train on LMP attacks with malicious clients $[5 : 5 : 50]$, and adapt to three cases with 20%, 30%, and 40% malicious clients in online adaptation, respectively. Similarly, when the *non-i.i.d.* level is between 0.1-1, the meta-SG will pre-train on LMP attacks with *non-i.i.d.* level $[0.1 : 0.1 : 1]$ and adapt to q= 0.5, 0.6, 0.7 in online adaptation. As illustrated in Figures 11(c) and 11(d), meta-SG reaches the highest model accuracy for all numbers of malicious clients and non-$i.i.d.$ levels under LMP.

**Importance of pre-training and online adaptation** As shown in Table 5, the pre-training is to derive defense policy rather than the model itself. Directly using those shifted data (root or generated) to train the FL model will result in model accuracy as low as 0.2-0.3 (0.4-0.5) for CIFAR-10 (MNIST) in our setting. Pre-training and online adaptation are indispensable in the proposed framework. Our experiments in Table 5 indicate that directly applying defense learned from pre-training w/o online adaptation and adaptation from randomly initialized defense policy w/o pre-training both fail to address malicious attacks, resulting in global model accuracy as low as 0.3-0.6 (0.1-0.4) on MNIST (CIFAR-10). In the absence of adaptation, meta policy itself falls short of the distribution shift between the simulated and the real environment. Likewise, the online adaptation fails to attain the desired defense policy without the pre-trained policy serving as a decent initialization.

**Biased/Limited root data** We evaluate the average model accuracy after 250 FL epochs under the meta-SG framework against the IPM attack, using root data with varying i.i.d. levels (as defined in the experiment setting section). Here, q = 0.1 (indicating the root data is i.i.d.) serves as our baseline meta-SG, as presented in the paper. We designate class 0 as the reference class. For instance, when q = 0.4, it indicates a 40% probability for each data labeled as class 0 within the root data, while the remaining 60% are distributed equally among the other classes. We observe that when q is as high as 0.7, there is one class (i.e., 3) missing in the root data. Although, through inverting methods in online adaptation, the defender can learn the missing data in the end, it suffered the slower adaptation compared with a good initial defense policy. In addition, we test the average model accuracy after 250 FL epochs under meta-SG against IPM attack using different numbers of root data (i.e., 100, 60, 20), where 100 root data is our original meta-SG setting in the rest of paper. We overserve that when number of root data is 20, two classes of data are missing (i.e., 1 and 5).

**Generalization to unseen adaptive attacks** We thoroughly search related works considering adaptive attacks in FL and find very limited works (with solid and lightweight open-source implementation) that can be used as our benchmark. As a result, we introduce two new benchmark adaptive attack methods in the testing stage as unseen adaptive attacks: (1) adaptive LMP![15], which requires access to normal clients' updates in each FL round, and (2) RL attack [31] restricted 1-dimensional action space (i.e., adaptive scalar factor) compared with the baseline 3-dimensional RL attack [31] showing in our paper. The defender in pre-training only interacts with the 3-dimensional RL attack. We test the average model accuracy after 250 FL epochs under meta-SG against different (unseen)

| Biased Level | q = 0.1 | q = 0.4 | q = 0.7 |
|---|---|---|---|
| Acc | 0.8951 | 0.8612 | 0.7572 |

(a) Ablation study of biased root data.

| Number of Root Data | 100 | 60 | 20 |
|---|---|---|---|
| Acc | 0.8951 | 0.8547 | 0.6902 |

(b) Ablation study of limited root data.

Table 6: Results of the average model accuracy on MNIST after 250 FL epochs under meta-SG against IPM attack using root data with (a) different i.i.d levels and (b) different numbers of root data. All random seeds are fixed and all other parameters are set as default.

| Acc/Bac | NormBound 0.2 | NormBound 0.1 | NormBound 0.05 |
|---|---|---|---|
| DBA | 0.6313/0.9987 | 0.5192/0.6994 | 0.3610/0.4392 |
| IPM+BFL | 0.6060/0.5123 | 0.4917/0.2104 | 0.3614/0.2253 |
| Acc/Bac | NeuroClip 10 | NeuroClip 6 | NeuroClip 1 |
| DBA | 0.6221/0.9974 | 0.6141/0.9984 | 0.2515/0.0002 |
| IPM+BFL | 0.1/0.0020 | 0.1/0 | 0.1/0 |

Table 7: Results of manually tuning norm threshold [57] and clipping range [62]. All other parameters are set as default and all random seeds are fixed.

adaptive attacks. What is interesting here is that meta-SG can achieve even better performance against unseen attacks.

| Attack Methods | Model Acc |
|---|---|
| 3-dimensional RL | 0.8652 |
| Adaptive LMP | 0.8692 |
| 1-dimensional RL | 0.8721 |

Table 8: Comparisons of average model accuracy after 250 FL rounds under different adaptive attacks on MNIST. All parameters are set as default and all random seeds are fixed.

# E   Algorithms

This section elaborates on meta-learning defense and meta-Stackelberg defense in equation meta-SE. To begin with, we first review the policy gradient method [58] in RL and its Monte-Carlo estimation. To simplify our exposition, we fix the attacker's policy $\phi$, and then the Markov game reduces to a single-agent MDP, where the optimal policy to be learned is the defender's $\theta$.

**Policy Gradient**   The idea of the policy gradient method is to apply gradient ascent to the value function $J_{\mathcal{D}}$. Following [58], we obtain $\nabla_\theta J_{\mathcal{D}} := \mathbb{E}_{\tau \sim q(\theta)}[g(\tau; \theta)]$, where $g(\tau; \theta) = \sum_{t=1}^{H} \nabla_\theta \log \pi(a_{\mathcal{D}}^t | s^t; \theta) R(\tau)$ and $R(\tau) = \sum_{t=1}^{H} \gamma^t r(s^t, a_{\mathcal{D}}^t)$. Note that for simplicity, we suppress the parameter $\phi, \xi$ in the trajectory distribution $q$, and instead view it as a function of $\theta$. In numerical implementations, the policy gradient $\nabla_\theta J_{\mathcal{D}}$ is replaced by its Monte-Carlo (MC) estimation using sample trajectory. Suppose a batch of trajectories $\{\tau_i\}_{i=1}^{N_b}$, and $N_b$ denotes the batch size, then the MC estimation is

$$\hat{\nabla}_\theta J_{\mathcal{D}}(\theta, \tau) := 1/N_b \sum_{\tau_i} g(\tau_i; \theta). \tag{E1}$$

The same deduction also holds for the attacker's problem when fixing the defense $\theta$.

**Meta-Learning FL Defense**   As discussed in Section 3, meta-learning-based defense (meta defense) mainly targets non-adaptive attack methods, where $\pi_{\mathcal{A}}(\cdot; \phi, \xi)$ is a pre-fixed attack strategy following some rulebook, such as IPM [68] and LMP [15]. In this case, the BSMG reduces to single-agent MDP for the defender, where the transition kernel is determined by the attack method. Mathematically, the meta-defense problem is given by

$$\max_{\theta, \Psi} \mathbb{E}_{\xi \sim Q(\cdot)}[J_{\mathcal{D}}(\Psi(\theta, \tau), \phi, \xi)]. \tag{E2}$$

Since the attack type is hidden from the defender, the adaptation mapping $\Psi$ is usually defined in a data-driven manner. For example, $\Psi(\theta, \tau)$ can be defined as a one-step stochastic gradient update with learning rate $\eta$: $\Psi(\theta, \tau) = \theta + \eta \hat{\nabla} J_{\mathcal{D}}(\tau_{\xi})$ [16] or a recurrent neural network in [13]. This work mainly focuses on gradient adaptation for the purpose of deriving theoretical guarantees in Appendix F.

With the one-step gradient adaptation, the meta-defense problem in equation E2 can be simplified as

$$\max_{\theta} \mathbb{E}_{\xi \sim Q(\cdot)} \mathbb{E}_{\tau \sim q(\theta)}[J_{\mathcal{D}}(\theta + \eta \hat{\nabla}_{\theta} J_{\mathcal{D}}(\tau), \phi, \xi)]. \tag{E3}$$

Recall that the attacker's strategy is pre-determined, $\phi, \xi$ can be viewed as fixed parameters, and hence, the distribution $q$ is a function of $\theta$. To apply the policy gradient method to equation E3, one needs an unbiased estimation of the gradient of the objective function in equation E3. Consider the gradient computation using the chain rule:

$$\nabla_{\theta} \mathbb{E}_{\tau \sim q(\theta)}[J_{\mathcal{D}}(\theta + \eta \hat{\nabla}_{\theta} J_{\mathcal{D}}(\tau), \phi, \xi)]$$
$$= \mathbb{E}_{\tau \sim q(\theta)} \{ \underbrace{\nabla_{\theta} J_{\mathcal{D}}(\theta + \eta \hat{\nabla}_{\theta} J_{\mathcal{D}}(\tau), \phi, \xi)(I + \eta \hat{\nabla}_{\theta}^2 J_D(\tau))}_{①}$$
$$+ \underbrace{J_{\mathcal{D}}(\theta + \eta \hat{\nabla}_{\theta} J_{\mathcal{D}}(\tau)) \nabla_{\theta} \sum_{t=1}^{H} \pi(a^t | s^t; \theta)}_{②} \}. \tag{E4}$$

The first term results from differentiating the integrand $J_{\mathcal{D}}(\theta + \eta \hat{\nabla}_{\theta} J_{\mathcal{D}}(\tau), \phi, \xi)$ (the expectation is taken as integration), while the second term is due to the differentiation of $q(\theta)$. One can see from the first term that the above gradient involves a Hessian $\hat{\nabla}^2 J_{\mathcal{D}}$, and its sample estimate is given by the following. For more details on this Hessian estimation, we refer the reader to [14].

$$\hat{\nabla}^2 J_{\mathcal{D}}(\tau) = \frac{1}{N_b} \sum_{i=1}^{N_b} [g(\tau_i; \theta) \nabla_{\theta} \log q(\tau_i; \theta)^{\mathsf{T}} + \nabla_{\theta} g(\tau_i; \theta)] \tag{E5}$$

Finally, to complete the sample estimate of $\nabla_{\theta} \mathbb{E}_{\tau \sim q(\theta)}[J_{\mathcal{D}}(\theta + \eta \hat{\nabla}_{\theta} J_{\mathcal{D}}(\tau), \phi, \xi)]$, one still needs to estimate $\nabla_{\theta} J_{\mathcal{D}}(\theta + \eta \hat{\nabla}_{\theta} J_{\mathcal{D}}(\tau), \phi, \xi)$ in the first term. To this end, we need to first collect a batch of sample trajectories $\tau'$ using the adapted policy $\theta' = \theta + \eta \hat{\nabla}_{\theta} J_D(\tau)$. Then, the policy gradient estimate of $\hat{\nabla}_{\theta} J_{\mathcal{D}}(\theta')$ proceeds as in equation E1. To sum up, constructing an unbiased estimate of equation E4 takes two rounds of sampling. The first round is under the meta policy $\theta$, which is used to estimate the Hessian equation E5 and to adapt the policy to $\theta'$. The second round aims to estimate the policy gradient $\nabla_{\theta} J_{\mathcal{D}}(\theta + \eta \hat{\nabla}_{\theta} J_{\mathcal{D}}(\tau), \phi, \xi)$ in the first term in equation E4.

In the experiment, we employ a first-order meta-learning algorithm called Reptile [43] to avoid the Hessian computation. The gist is to simply ignore the chain rule and update the policy using the gradient $\nabla_{\theta} J_{\mathcal{D}}(\theta', \phi, \xi)|_{\theta' = \theta + \eta \hat{\nabla}_{\theta} J_{\mathcal{D}}(\tau)}$. Naturally, without the Hessian term, the gradient in this update is biased, yet it still points to the ascent direction as argued in [43], leading to effective meta policy. The advantage of Reptile is more evident in multi-step gradient adaptation. Consider a $l$-step gradient adaptation, the chain rule computation inevitably involves multiple Hessian terms (each gradient step brings a Hessian term) as shown in [14]. In contrast, Reptile only requires first-order information, and the meta-learning algorithm ($l$-step adaptation) is given by Algorithm 2.

**Meta-Stackelberg Learning**   Recall that in meta-SE, the attacker's policy $\phi_{\xi}^*$ is not pre-fixed. Instead, it is the best response to the defender's adapted policy as shown in equation meta-SE. To

---

**Algorithm 2** Reptile Meta-Reinforcement Learning with $l$-step adaptation

---

1: **Input:** the type distribution $Q(\xi)$, step size parameters $\kappa, \eta$
2: **Output:** $\theta^T$
3: randomly initialize $\theta^0$
4: **for** iteration $t = 1$ to $T$ **do**
5:     Sample a batch $\hat{\Xi}$ of $K$ attack types from $Q(\xi)$;
6:     **for** each $\xi \in \hat{\Xi}$ **do**
7:         $\theta_\xi^t(0) \leftarrow \theta^t$
8:         **for** $k = 0$ to $l - 1$ **do**
9:             Sample a batch trajectories $\tau$ of the horizon length $H$ under $\theta_\xi^t(k)$;
10:             Evaluate $\hat{\nabla}_\theta J_{\mathcal{D}}(\theta_\xi^t(k), \tau)$ using MC in equation E1;
11:             $\theta_\xi^t(k+1) \leftarrow \theta_\xi^t(k) + \kappa \hat{\nabla}_\theta J_{\mathcal{D}}(\theta^t, \tau)$
12:         **end for**
13:     **end for**
14:     Update $\theta^{t+1} \leftarrow \theta^t + 1/K \sum_{\xi \in \hat{\Xi}} (\theta_\xi^t(l) - \theta^t)$;
15: **end for**

---

944   obtain this best response, one needs alternative training: fixing the defense policy, and applying
945   gradient ascent to the attacker's problem until convergence. It should be noted that the proposed
946   meta-SL utilizes the unbiased gradient estimation in equation E5, which paves the way for theoretical
947   analysis in Appendix F. Yet, we turn to the Reptile to speed up pre-straining in the experiments. We
948   present both algorithms in Algorithm 3, and only consider one-step adaptation for simplicity. The
multi-step version is a straightforward extension of Algorithm 3.

---

**Algorithm 3** (Reptile) Meta-Stackelberg Learning with one-step adaptation

---

1: **Input:** the type distribution $Q(\xi)$, initial defense meta policy $\theta^0$, pre-trained attack policies $\{\phi_\xi^0\}_{\xi \in \Xi}$, step size parameters $\kappa_{\mathcal{D}}, \kappa_{\mathcal{A}}, \eta$, and iterations numbers $N_{\mathcal{A}}, N_{\mathcal{D}}$;
2: **Output:** $\theta^{N_{\mathcal{D}}}$
3: **for** iteration $t = 0$ to $N_{\mathcal{D}} - 1$ **do**
4:     Sample a batch $\hat{\Xi}$ of $K$ attack types from $Q(\xi)$;
5:     **for** each $\xi \in \hat{\Xi}$ **do**
6:         Sample a batch of trajectories using $\phi^t$ and $\phi_\xi^t$;
7:         Evaluate $\hat{\nabla}_\theta J_D(\theta^t, \phi_\xi^t, \xi)$ using equation E1;
8:         Perform one-step adaptation $\theta_\xi^t \leftarrow \theta^t + \eta \hat{\nabla}_\theta J_D(\theta_\xi^t(k), \phi_\xi^t, \xi)$;
9:         $\phi_\xi^t(0) \leftarrow \phi_\xi^t$;
10:         **for** $k = 0, \ldots, N_{\mathcal{A}} - 1$ **do**
11:             Sample a batch of trajectories using $\theta_\xi^t$ and $\phi_\xi^t(k)$;
12:             $\phi_\xi^t(k+1) \leftarrow \phi_\xi^t(k) + \kappa_{\mathcal{A}} \hat{\nabla}_\phi J_{\mathcal{A}}(\theta_\xi^t, \phi_\xi^t(k), \xi)$;
13:         **end for**
14:         **if** Reptile **then**
15:             Sample a batch of trajectories using $\theta_\xi^t$ and $\phi_\xi^t(N_{\mathcal{A}})$;
16:             Evaluate $\hat{\nabla} J_D(\xi) := \hat{\nabla}_\theta J_{\mathcal{D}}(\theta, \phi_\xi^t(N_{\mathcal{A}}), \xi)|_{\theta = \theta_\xi^t}$ using equation E1;
17:         **else**
18:             Sample a batch of trajectories using $\theta^t$ and $\phi_\xi^t(N_{\mathcal{A}})$;
19:             Evaluate the Hessian using equation E5;
20:             Sample a batch of trajectories using $\theta_\xi^t$ and $\phi_\xi^t(N_{\mathcal{A}})$;
21:             Evaluate $\hat{\nabla} J_D(\xi) := \hat{\nabla}_\theta J_{\mathcal{D}}(\theta_\xi^t, \phi_\xi^t(N_{\mathcal{A}}), \xi)$ using equation E4;
22:         **end if**
23:         $\bar{\theta}_\xi^t \leftarrow \theta^t + \kappa_{\mathcal{D}} \hat{\nabla} J_D(\xi)$;
24:     **end for**
25:     $\theta^{t+1} \leftarrow \theta^t + 1/K \sum_{\xi \sim \hat{\Xi}} (\bar{\theta}_\xi^t - \theta_t), \phi_\xi^{t+1} \leftarrow \phi_\xi^t(N_{\mathcal{A}})$;
26: **end for**

---

 # F   Theoretical Results

 ## F.1   Existence of Meta-SG

 **Theorem F.1.** *Under the conditions that $\Theta$ and $\Phi$ are compact and convex, the meta-SG admits at*
 *least one meta-FOSE.*

 *Proof.* Clearly, $\Theta \times \Phi^{|\Xi|}$ is compact and convex, let $\phi \in \Phi^{|\Xi|}, \phi_\xi \in \Phi$ be the (type-aggregated)
 attacker's strategy, since the consider twice continuously differentiable utility functions $\ell_{\mathcal{D}}(\theta, \phi) :=$
 $\mathbb{E}_{\xi \sim Q} \mathcal{L}_{\mathcal{D}}(\theta, \phi_\xi, \xi)$ and $\ell_\xi(\theta, \phi) := \mathcal{L}_{\mathcal{A}}(\theta, \phi_\xi, \xi)$ for all $\xi \in \Xi$. Then, there exists a constant $\gamma_c > 0$,
 such that the auxiliary utility functions:

$$\tilde{\ell}_{\mathcal{D}}(\theta; (\theta', \phi')) \equiv \ell_{\mathcal{D}}(\theta, \phi) - \frac{\gamma_c}{2}\|\theta - \theta'\|^2$$
$$\tilde{\ell}_\xi(\phi_\xi; (\theta', \phi') \equiv \ell_\xi(\theta', (\phi_\xi, \phi'_{-\xi})) - \frac{\gamma_c}{2}\|\phi_\xi - \phi'_\xi\|^2 \quad \forall \xi \in \Xi \tag{F6}$$

 are $\gamma_c$-strongly concave in spaces $\theta \in \Theta, \phi_\xi \in \Phi$ for all $\xi \in \Xi$, respectively for any fixed $(\theta', \phi') \in$
 $\Theta \times \Phi^{|\Xi|}$.

 Define the self-map $h : \Theta \times \Phi^{|\Xi|} \to \Theta \times \Phi^{|\Xi|}$ with $h(\theta', \phi') \equiv (\bar{\theta}(\theta', \phi'), \bar{\phi}(\theta', \phi'))$, where

$$\bar{\theta}(\theta', \phi') = \arg\max_{\theta \in \Theta} \tilde{\ell}_{\mathcal{D}}(\theta, \phi'), \qquad \bar{\phi}_\xi(\theta', \phi') = \arg\max_{\phi_\xi \in \Phi} \tilde{\ell}_\xi(\theta', (\phi_\xi, \phi'_{-\xi})).$$

 Due to compactness, $h$ is well-defined. By strong concavity of $\tilde{\ell}_{\mathcal{D}}(\cdot; (\theta', \phi'))$ and $\tilde{\ell}_\xi(\cdot; (\theta', \phi'))$, it
 follows that $\bar{\theta}, \bar{\phi}$ are continuous self-mapping from $\Theta \times \Phi^{|\Xi|}$ to itself. By Brouwer's fixed point
 theorem, there exists at least one $(\theta^*, \phi^*) \in \Theta \times \Phi^{|\Xi|}$ such that $h(\theta^*, \phi^*) = (\theta^*, \phi^*)$. Then, one can
 verify that $(\theta^*, \phi^*)$ is a meta-FOSE of the meta-SG with utility function $\ell_{\mathcal{D}}$ and $\ell_\xi, \xi \in \Xi$, in view of
 the following inequality

$$\langle \nabla_\theta \tilde{\ell}_{\mathcal{D}}(\theta^*; (\theta^*, \phi^*)), \theta - \theta^* \rangle = \langle \nabla_\theta \ell_{\mathcal{D}}(\theta^*, \phi^*), \theta - \theta^* \rangle$$
$$\langle \nabla_{\phi_\xi} \tilde{\ell}_\xi(\theta^*; (\theta^*, \phi^*)), \phi_\xi - \phi_\xi^* \rangle = \langle \nabla_{\phi_\xi} \ell_\xi(\theta^*, \phi^*), \phi_\xi - \phi_\xi^* \rangle,$$

 therefore, the equilibrium conditions for meta-SG with utility functions $\tilde{\ell}_{\mathcal{D}}$ and $\{\tilde{\ell}_\xi\}_{\xi \in \Xi}$ are the same
 as with utility functions $\ell_{\mathcal{D}}$ and $\{\ell_\xi\}_{\xi \in \Xi}$, hence the claim follows. $\qquad \square$

 ## F.2   Proofs: Non-Asymptotic Analysis

 In the sequel, we make the following smoothness assumptions for every attack type $\xi \in \Xi$. In
 addition, we assume, for analytical simplicity, that all types of attackers are unconstrained, i.e., $\Phi$ is
 the Euclidean space with proper finite dimension.

 **Assumption F.2** (($\xi$-wise) Lipschitz smoothness)**.** The functions $\mathcal{L}_{\mathcal{D}}$ and $\mathcal{L}_{\mathcal{A}}$ are continuously
 diffrentiable in both $\theta$ and $\phi$. Furthermore, there exists constants $L_{11}, L_{12}, L_{21}$, and $L_{22}$ such that
 for all $\theta, \theta_1, \theta_2 \in \Theta$ and $\phi, \phi_1, \phi_2 \in \Phi$, we have, for any $\xi \in \Xi$,

$$\|\nabla_\theta \mathcal{L}_{\mathcal{D}}(\theta_1, \phi, \xi) - \nabla_\theta \mathcal{L}_{\mathcal{D}}(\theta_2, \phi, \xi)\| \le L_{11} \|\theta_1 - \theta_2\| \tag{F7}$$
$$\|\nabla_\phi \mathcal{L}_{\mathcal{D}}(\theta, \phi_1, \xi) - \nabla_\phi \mathcal{L}_{\mathcal{D}}(\theta, \phi_2, \xi)\| \le L_{22} \|\phi_1 - \phi_2\| \tag{F8}$$
$$\|\nabla_\theta \mathcal{L}_{\mathcal{D}}(\theta, \phi_1, \xi) - \nabla_\theta \mathcal{L}_{\mathcal{D}}(\theta, \phi_2, \xi)\| \le L_{12} \|\phi_1 - \phi_2\| \tag{F9}$$
$$\|\nabla_\phi \mathcal{L}_{\mathcal{D}}(\theta_1, \phi, \xi) - \nabla_\phi \mathcal{L}_{\mathcal{D}}(\theta_2, \phi, \xi)\| \le L_{12} \|\theta_1 - \theta_2\| \tag{F10}$$
$$\|\nabla_\phi \mathcal{L}_{\mathcal{A}}(\theta, \phi_1, \xi) - \nabla_\phi \mathcal{L}_{\mathcal{A}}(\theta, \phi_2, \xi)\| \le L_{21} \|\phi_1 - \phi_2\|. \tag{F11}$$

 We also make the following strict-competitiveness assumption. This notion can be treated as a
 generalization of zero-sum games: if one joint action $(a_{\mathcal{D}}, a_{\mathcal{A}})$ leads to payoff increases for one
 player, it must decrease the other's payoff.

 **Assumption F.3** (Strict-Competitiveness)**.** The BSMG is strictly competitive, i.e., there exist con-
 stants $c < 0, d$ such that $\forall \xi \in \Xi, s \in S, a_{\mathcal{D}}, a_{\mathcal{A}} \in A_{\mathcal{D}} \times A_\xi, r_{\mathcal{D}}(s, a_{\mathcal{D}}, a_{\mathcal{A}}) = c r_{\mathcal{A}}(s, a_{\mathcal{D}}, a_{\mathcal{A}}) + d.$

In adversarial FL, the untargeted attack naturally makes the game zero-sum (hence, SC). The purpose of introducing Assumption F.3 is to establish the Danskin-type result [3] for the Stackelberg game with nonconvex value functions (see Lemma F.5), which spares us from the Hessian inversion.

**Lemma F.4** (Implicit Function Theorem (IFT) for Meta-SG). *Suppose for $(\bar{\theta}, \bar{\phi}) \in \Theta \times \Phi^{|\Xi|}$, $\xi \in \Xi$ we have $\nabla_\phi \mathcal{L}_\mathcal{A}(\bar{\theta}, \bar{\phi}, \xi) = 0$ the Hessian $\nabla^2_\phi \mathcal{L}_\mathcal{A}(\bar{\theta}, \bar{\phi}, \xi)$ is non-singular. Then, there exists a neighborhood $B_\varepsilon(\bar{\theta}), \varepsilon > 0$ centered around $\bar{\theta}$ and a $C^1$-function $\phi(\cdot) : B_\varepsilon(\bar{\theta}) \to \Phi^{|\Xi|}$ such that near $(\bar{\theta}, \bar{\phi})$ the solution set $\{(\theta, \phi) \in \Theta \times \Phi^{|\Xi|} : \nabla_\phi \mathcal{L}_\mathcal{A}(\theta, \phi, \xi) = 0\}$ is a $C^1$-manifold locally near $(\bar{\theta}, \bar{\phi})$. The gradient $\nabla_\theta \phi(\theta)$ is given by $-(\nabla^2_\phi \mathcal{L}_\mathcal{A}(\theta, \phi, \xi))^{-1} \nabla^2_{\phi\theta} \mathcal{L}_\mathcal{A}(\theta, \phi, \xi)$.*

**Lemma F.5.** *Under assumptions F.2, 3.2, there exists $\{\phi_\xi : \phi_\xi \in \arg\max_\phi \mathcal{L}_\mathcal{A}(\theta, \phi, \xi)\}_{\xi \in \Xi}$, such that*

$$\nabla_\theta V(\theta) = \nabla_\theta \mathbb{E}_{\xi \sim Q, \tau \sim q} J_\mathcal{D}(\theta + \eta \hat{\nabla}_\theta J_\mathcal{D}(\tau), \phi_\xi, \xi).$$

*Moreover, the function $V(\theta)$ is L-Lipschitz-smooth, where $L = L_{11} + \frac{L_{12}L_{21}}{\mu}$*

$$\|\nabla_\theta V(\theta_1) - \nabla_\theta V(\theta_2)\| \le L \|\theta_1 - \theta_2\|.$$

*Proof of Lemma F.5.* First, we show that for any $\theta_1, \theta_2 \in \Theta, \xi \in \Xi$, and $\phi_1 \in \arg\max_\phi \mathcal{L}_\mathcal{A}(\theta_1, \phi, \xi)$, there exists $\phi_2 \in \arg\max_\phi \mathcal{L}_\mathcal{A}(\theta_2, \phi, \xi)$ such that $\|\phi_1 - \phi_2\| \le \frac{L_{12}}{\mu} \|\theta_1 - \theta_2\|$. Indeed, based on smoothness assumption equation F11 and equation F10,

$$\|\nabla_\phi \mathcal{L}_\mathcal{A}(\theta_1, \phi_1, \xi) - \nabla_\phi \mathcal{L}_\mathcal{A}(\theta_2, \phi_1, \xi)\| \le L_{21} \|\theta_1 - \theta_2\|,$$
$$\|\nabla_\phi \mathcal{L}_\mathcal{D}(\theta_1, \phi_1, \xi) - \nabla_\phi \mathcal{L}_\mathcal{D}(\theta_2, \phi_1, \xi)\| \le L_{12} \|\theta_1 - \theta_2\|.$$

Since $\phi_2 \in \arg\max_\phi \mathcal{L}_\mathcal{A}(\theta_2, \phi, \xi), \nabla_\phi \mathcal{L}_\mathcal{A}(\theta_2, \phi_2, \xi) = 0$. Apply PL condition to $\nabla_\phi \mathcal{L}_\mathcal{A}(\theta, \phi_2, \xi)$,

$$\max_\phi \mathcal{L}_\mathcal{A}(\theta_1, \phi, \xi) - \mathcal{L}_\mathcal{A}(\theta_1, \phi_2, \xi) \le \frac{1}{2\mu} \|\nabla_\phi \mathcal{L}_\mathcal{A}(\theta_1, \phi_2, \xi)\|^2$$
$$= \frac{1}{2\mu} \|\nabla_\phi \mathcal{L}_\mathcal{A}(\theta_1, \phi_2, \xi) - \nabla_\phi \mathcal{L}_\mathcal{A}(\theta_2, \phi_2, \xi)\|^2$$
$$\le \frac{L_{21}^2}{2\mu} \|\theta_1 - \theta_2\|^2 \qquad \text{by equation F11.}$$

Since PL condition implies quadratic growth, we also have

$$\mathcal{L}_\mathcal{A}(\theta_1, \phi_1, \xi) - \mathcal{L}_\mathcal{A}(\theta_1, \phi_2, \xi) \ge \frac{\mu}{2} \|\phi_1 - \phi_2\|^2.$$

Combining the two inequalities above we obtain the Lipschitz stability for $\phi_\xi^*(\cdot)$, i.e.,

$$\|\phi_1 - \phi_2\| \le \frac{L_{21}}{\mu} \|\theta_1 - \theta_2\|.$$

Second, show that $\nabla_\theta V(\theta)$ can be directly evaluated at $\{\phi_\xi^*\}_{\xi \in \Xi}$. Inspired by Danskin's theorem, we first made the following argument, consider the definition of directional derivative. Let $\ell(\theta, \phi) := \nabla_\theta \mathbb{E}_{\xi, \tau} J_\mathcal{D}(\theta + \eta \hat{\nabla} J_\mathcal{D}(\tau), \xi)$. For a constant $\tau$ and an arbitrary direction $d$,

$$\ell(\theta + \tau d, \phi^*(\theta + \tau d)) - \ell(\theta, \phi^*(\theta)))$$
$$= \ell(\theta + \tau d, \phi^*(\theta + \tau d)) - \ell(\theta + \tau d, \phi^*(\theta)) + \ell(\theta + \tau d, \phi^*(\theta)) - \ell(\theta, \phi^*(\theta))$$
$$= \nabla_\phi \ell(\theta + \tau d, \phi^*(\theta))^\top \underbrace{[\phi^*(\theta + \tau d) - \phi^*(\theta))]}_{\Delta\phi} + o(\Delta\phi^2)$$
$$+ \tau \nabla_\theta \ell(\theta, \phi^*(\theta))^T d + o(d^2).$$

Hence, a sufficient condition for the first equation is $\nabla_\phi \ell(\theta + \tau d, \phi^*(\theta)) = 0$, meaning that $\ell_D(\theta, \phi)$ and $\mathcal{L}_\mathcal{A}(\theta, \phi, \xi)$ share the first-order stationarity at every $\phi$ when fixing $\theta$. Indeed, by Lemma F.4, we have, the gradient is locally determined by

$$\nabla_\theta V = \mathbb{E}_{\xi \sim Q}[\nabla_\theta \mathcal{L}_\mathcal{D}(\theta, \phi_\xi, \xi) + (\nabla_\theta \phi_\xi(\theta))^\top \nabla_\phi \mathcal{L}_\mathcal{D}(\theta, \phi_\xi, \xi)]$$
$$= \mathbb{E}_{\xi \sim Q}\left[\nabla_\theta \mathcal{L}_\mathcal{D}(\theta, \phi_\xi, \xi) - [(\nabla^2_\phi \mathcal{L}_\mathcal{A}(\theta, \phi, \xi))^{-1} \nabla^2_{\phi\theta} \mathcal{L}_\mathcal{A}(\theta, \phi, \xi)]^\top \nabla_\phi \mathcal{L}_\mathcal{D}(\theta, \phi_\xi, \xi)\right].$$

Given a trajectory $\tau := (s^1, a_{\mathcal{D}}^t, a_{\mathcal{A}}^t, \ldots, a_{\mathcal{D}}^H, a_{\mathcal{A}}^H, s^{H+1})$, let $R_{\mathcal{D}}(\tau, \xi) := \sum_{t=1}^{H} \gamma^{t-1} r_{\mathcal{D}}(s_t, a_t, \xi)$ and $R_{\mathcal{D}}(\tau, \xi) := \sum_{t=1}^{H} \gamma^{t-1} r_{\mathcal{D}}(s_t, a_t, \xi)$. Denote by $\mu(\tau; \theta, \phi)$ the trajectory distribution, that the log probability of $\mu$ is given by

$$\log \mu(\tau; \theta, \phi) = \sum_{t=1}^{H} (\log \pi_{\mathcal{D}}(a_{\mathcal{D}}^t | s^t; \theta + \eta \hat{\nabla}_\theta J_{\mathcal{D}}(\tau)) + \log \pi_{\mathcal{A}}(a_{\mathcal{A}}^t | s^t; \phi) + \log P(s^{t+1} | a_{\mathcal{D}}^t, a_{\mathcal{A}}^t, s^t)$$

According to the policy gradient theorem, we have

$$\nabla_\phi \mathcal{L}_{\mathcal{D}}(\theta, \phi, \xi) = \mathbb{E}_\mu [R_{\mathcal{D}}(\tau, \xi) \sum_{t=1}^{H} \nabla_\phi \log(\pi_{\mathcal{A}}(a_{\mathcal{A}}^t | s^t; \phi))],$$

$$\nabla_\phi \mathcal{L}_{\mathcal{A}}(\theta, \phi, \xi) = \mathbb{E}_\mu [R_{\mathcal{A}}(\tau, \xi) \sum_{t=1}^{H} \nabla_\phi \log(\pi_{\mathcal{A}}(a_{\mathcal{A}}^t | s^t; \phi))].$$

By SC Assumption F.3, when $\nabla_\phi \mathcal{L}_{\mathcal{A}}(\theta, \phi, \xi) = 0$, there exists $c < 0$, $d$, such that $\nabla_\phi \mathcal{L}_{\mathcal{D}}(\theta, \phi, \xi) = \mathbb{E}_\mu [c R_{\mathcal{A}}(\tau, \xi) \sum_{t=1}^{H} \nabla_\phi \log(\pi_{\mathcal{A}}(a_{\mathcal{A}}^t | s^t; \phi))] + \mathbb{E}_\mu [\sum_{t=1}^{H} \gamma^{t-1} d \sum_{t=1}^{H} \nabla_\phi \log(\pi_{\mathcal{A}}(a_{\mathcal{A}}^t | s^t; \phi))] = 0$. Hence $\nabla_\theta V = \mathbb{E}_{\xi \sim Q}[\nabla_\theta \mathcal{L}_{\mathcal{D}}(\theta, \phi_\xi, \xi)]$.

Third, $V(\theta)$ is also Lipschitz smooth. As we notice that, $\ell_{\mathcal{D}}$ is Lipschitz smooth since $\mathbb{E}_{\xi \sim Q}$ is a linear operator, we have,

$$\begin{aligned}
&\|\nabla_\theta V(\theta_1) - \nabla_\theta V(\theta_2)\| \\
&\leq \|\nabla_\theta \mathbb{E}_{\xi \sim Q} \mathcal{L}_{\mathcal{D}}(\theta_1, \phi_1, \xi) - \nabla_\theta \mathbb{E}_{\xi \sim Q} \mathcal{L}_{\mathcal{D}}(\theta_2, \phi_2, \xi)\| \\
&= \|\nabla_\theta \ell_{\mathcal{D}}(\theta_1, \phi_1) - \nabla_\theta \ell_{\mathcal{D}}(\theta_2, \phi_1) + \nabla_\theta \ell_{\mathcal{D}}(\theta_2, \phi_1) - \nabla_\theta \ell_{\mathcal{D}}(\theta_2, \phi_2)\| \\
&\leq \|\nabla_\theta \ell_{\mathcal{D}}(\theta_1, \phi_1) - \nabla_\theta \ell_{\mathcal{D}}(\theta_2, \phi_1)\| + \|\nabla_\theta \ell_{\mathcal{D}}(\theta_2, \phi_1) - \nabla_\theta \ell_{\mathcal{D}}(\theta_2, \phi_2)\| \\
&\leq L_{11} \|\theta_1 - \theta_2\| + L_{12} \|\phi_1 - \phi_2\| \\
&\leq (L_{11} + \frac{L_{12} L_{21}}{\mu}) \|\theta_1 - \theta_2\|,
\end{aligned}$$

which implies the Lipschitz constant $L = L_{11} + \frac{L_{12} L_{21}}{\mu}$. $\qquad \square$

It is impossible to present the convergence theory without the assistance of some standard assumptions in batch reinforcement learning, of which the justification can be found in [14]. We also require some additional information about the parameter space and function structure. These assumptions are all stated in Assumption F.6.

**Assumption F.6.**

(a) The policy gradients are bounded, $\|\nabla_\theta \mathcal{L}_{\mathcal{D}}(\theta, \phi, \xi)\| \leq G^2$, $\|\nabla_\phi \mathcal{L}_{\mathcal{A}}(\theta, \phi, \xi)\| \leq G^2$ for all $\theta, \phi \in \Theta \times \Phi$ and $\xi \in \Xi$.

(b) The policy gradient estimations are unbiased, i.e.,

$$\mathbb{E}[\hat{\nabla}_\phi J_{\mathcal{A}}(\theta^t, \phi_\xi^t, \xi) - \nabla_\phi J_{\mathcal{A}}(\theta^t, \phi_\xi^t, \xi)] = 0$$

(c) The variances for the stochastic gradients are bounded, i.e., for all $\theta^t, \phi_\xi^t, \xi$,

$$\mathbb{E}[\|\hat{\nabla}_\phi J_{\mathcal{A}}(\theta^t, \phi_\xi^t, \xi) - \nabla_\phi J_{\mathcal{A}}(\theta^t, \phi_\xi^t, \xi)\|^2] \leq \frac{\sigma^2}{N_b}.$$

$$\mathbb{E}[\|\hat{\nabla}_\phi J_{\mathcal{D}}(\theta^t, \phi_\xi^t, \xi) - \nabla_\theta J_{\mathcal{D}}(\theta^t, \phi_\xi^t, \xi)\|^2] \leq \frac{\sigma^2}{N_b}.$$

(d) The parameter space $\Theta$ has diameter $D_\Theta := \sup_{\theta_1, \theta_2 \in \Theta} \|\theta_1 - \theta_2\|$; the initialization $\theta^0$ admits at most $D_V$ function gap, i.e., $D_V := \max_{\theta \in \Theta} V(\theta) - V(\theta^0)$.

(e) It holds that the parameters satisfy $0 < \mu < -c L_{22}$.

Equipped with Assumption F.6 we are able to unfold our main result Theorem 3.3, before which we show in Lemma F.7 that $\phi_\xi^*$ can be efficiently approximated by the inner loop in the sense that $\nabla_\theta \mathbb{E}_{\xi \sim Q} \mathcal{L}_{\mathcal{D}}(\theta^t, \phi_\xi^t(N_{\mathcal{A}}), \xi) \approx \nabla_\theta V(\theta^t)$, where $\phi_\xi^t(N_{\mathcal{A}})$ is the last iterate output of the attacker policy.

**Lemma F.7.** *Under Assumption F.6, 3.2, F.3, and F.2, let $\rho := 1 + \frac{\mu}{cL_{22}} \in (0,1)$, $\bar{L} = \max\{L_{11}, L_{12}, L_{22}, L_{21}, V_\infty\}$ where $V_\infty := \max\{\max\|\nabla V(\theta)\|, 1\}$. For all $\varepsilon > 0$, if the attacker learning iteration $N_{\mathcal{A}}$ and batch size $N_b$ are large enough such that*

$$N_{\mathcal{A}} \geq \frac{1}{\log \rho^{-1}} \log \frac{32 D_V^2 (2V_\infty + LD_\Theta)^4 \bar{L}|c|G^2}{L^2 \mu^2 \varepsilon^4}$$

$$N_b \geq \frac{32\mu L_{21}^2 D_V^2 (2V_\infty + LD_\Theta)^4}{|c|L_{22}^2 \sigma^2 \bar{L} L \varepsilon^4},$$

*then, for $z_t := \nabla_\theta \mathbb{E}_{\xi \sim Q} \mathcal{L}_{\mathcal{D}}(\theta^t, \phi_\xi^t(N_{\mathcal{A}}), \xi) - \nabla_\theta V(\theta^t)$,*

$$\mathbb{E}[\|z_t\|] \leq \frac{L\varepsilon^2}{4D_V(2V_\infty + LD_\Theta)^2},$$

*and*

$$\mathbb{E}[\|\nabla_\phi \mathcal{L}_{\mathcal{A}}(\theta^t, \phi_\xi^t(N), \xi)\|] \leq \varepsilon.$$

*Proof of Lemma F.7.* Fixing a $\xi \in \Xi$, due to Lipschitz smoothness,
$$\mathcal{L}_{\mathcal{D}}(\theta^t, \phi_\xi^t(N), \xi) - \mathcal{L}_{\mathcal{D}}(\theta^t, \phi_\xi^t(N-1), \xi)$$
$$\leq \langle \nabla_\phi \mathcal{L}_{\mathcal{D}}(\theta^t, \phi_\xi^t(N-1), \xi), \phi_\xi^t(N) - \phi_\xi^t(N-1) \rangle + \frac{L_{22}}{2} \|\phi_\xi^t(N) - \phi_\xi^t(N-1)\|^2.$$

The inner loop updating rule ensures that when $\kappa_{\mathcal{A}} = \frac{1}{L_{21}}$, $\phi_\xi^t(N) - \phi_\xi^t(N-1) = \frac{1}{L_{21}} \hat{\nabla}_\phi J_{\mathcal{A}}(\theta_\xi^t, \phi_\xi^t(N-1), \xi)$. Plugging it into the inequality, we arrive at
$$\mathcal{L}_{\mathcal{D}}(\theta^t, \phi_\xi^t(N), \xi) - \mathcal{L}_{\mathcal{D}}(\theta^t, \phi_\xi^t(N-1), \xi)$$
$$\leq \frac{1}{L_{21}} \langle \nabla_\phi \mathcal{L}_{\mathcal{D}}(\theta^t, \phi_\xi^t(N-1), \xi), \hat{\nabla}_\phi J_{\mathcal{A}}(\theta_\xi^t, \phi_\xi^t(N-1), \xi) \rangle + \frac{L_{22}}{2L_{21}^2} \|\hat{\nabla}_\phi J_{\mathcal{A}}(\theta_\xi^t, \phi_\xi^t(N-1), \xi)\|^2.$$

Therefore, we let $(\mathcal{F}_n^t)_{0 \leq n \leq N}$ be the filtration generated by $\sigma(\{\phi_\xi^t(\tau)\}_{\xi \in \Xi} | \tau \leq n)$ and take conditional expectations on $\mathcal{F}_n^t$:
$$\mathbb{E}[V(\theta^t) - \ell_{\mathcal{D}}(\theta^t, \phi^t(N))|\mathcal{F}_{N-1}^t] \leq V(\theta^t) - \ell_{\mathcal{D}}(\theta^t, \phi^t(N-1))$$
$$\mathbb{E}_\xi \left[ \frac{1}{L_{21}} \langle \nabla_\phi \mathcal{L}_{\mathcal{D}}, \nabla_\phi J_{\mathcal{A}}(\theta_\xi^t, \phi_\xi^t(N-1), \xi) \rangle + \frac{L_{22}}{2L_{21}^2} \|\hat{\nabla}_\phi J_{\mathcal{A}}(\theta_\xi^t, \phi_\xi^t(N-1), \xi)\|^2 \right].$$

By variance-bias decomposition, and Assumption F.6 (b) and (c),
$$\mathbb{E}[\|\hat{\nabla}_\phi J_{\mathcal{A}}(\theta_\xi^t, \phi_\xi^t(N-1), \xi)\|^2 | \mathcal{F}_{N-1}^t]$$
$$= \mathbb{E}[\|\hat{\nabla}_\phi J_{\mathcal{A}}(\theta_\xi^t, \phi_\xi^t(N-1), \xi) - \nabla_\phi J_{\mathcal{A}}(\theta_\xi^t, \phi_\xi^t(N-1), \xi) + \nabla_\phi J_{\mathcal{A}}(\theta_\xi^t, \phi_\xi^t(N-1), \xi)\|^2 | \mathcal{F}_{N-1}^t]$$
$$= \mathbb{E}[\|(\hat{\nabla}_\phi - \nabla_\phi) J_{\mathcal{A}}(\theta_\xi^t, \phi_\xi^t(N-1), \xi)\|^2 | \mathcal{F}_{N-1}^t] + \mathbb{E}[\|\nabla_\phi J_{\mathcal{A}}(\theta_\xi^t, \phi_\xi^t(N-1), \xi)\|^2 | \mathcal{F}_{N-1}^t]$$
$$+ \mathbb{E}[2\langle (\hat{\nabla}_\phi - \nabla_\phi) J_{\mathcal{A}}(\theta_\xi^t, \phi_\xi^t(N-1), \xi), \nabla_\phi J_{\mathcal{A}}(\theta_\xi^t, \phi_\xi^t(N-1), \xi) \rangle | \mathcal{F}_{N-1}^t]$$
$$\leq \frac{\sigma^2}{N_b} + \|\nabla_\phi J_{\mathcal{A}}(\theta_\xi^t, \phi_\xi^t(N-1), \xi)\|^2.$$

Applying the PL condition (Assumption 3.2), and Assumption F.6 (a) we obtain
$$\mathbb{E}[V(\theta^t) - \ell_{\mathcal{D}}(\theta, \phi^t(N))|\phi^{N-1}] - V(\theta^t) - \ell_{\mathcal{D}}(\theta, \phi^t(N-1))$$
$$\leq \mathbb{E}_\xi \left[ \frac{1}{L_{21}} \langle \nabla_\phi \mathcal{L}_{\mathcal{D}}, \nabla_\phi \mathcal{L}_{\mathcal{A}}(\theta^t, \phi_\xi^t(N-1), \xi) \rangle + \frac{L_{22}}{2L_{21}^2} (\frac{\sigma^2}{N_b} + \|\nabla_\phi \mathcal{L}_{\mathcal{A}}(\theta^t, \phi_\xi^t(N-1), \xi)\|^2) \right]$$
$$= \mathbb{E}_\xi \left[ -\frac{1}{2L_{22}} \|\nabla_\phi \mathcal{L}_{\mathcal{D}}\|^2 + \frac{1}{2L_{22}} \|\nabla_\phi (\mathcal{L}_{\mathcal{D}} + \frac{L_{22}}{L_{21}} \mathcal{L}_{\mathcal{A}})(\theta^t, \phi_\xi^t(N-1), \xi)\|^2 + \frac{L_{22}\sigma^2}{2L_{21}^2 N_b} \right]$$
$$\leq \frac{\mu}{cL_{21}} (\max_\phi \ell_{\mathcal{D}}(\theta^t, \phi) - \ell_{\mathcal{D}}(\theta^t, \phi^t(N-1))) + \frac{L_{22}\sigma^2}{2L_{21}^2 N_b},$$

rearranging the terms yields

$$\mathbb{E}[V(\theta^t) - \ell_{\mathcal{D}}(\theta^t, \phi^t(N))|\mathcal{F}_n^t] \le \rho(V(\theta^t) - \ell_{\mathcal{D}}(\theta^t, \phi^t(N-1))) + \frac{L_{22}\sigma^2}{2L_{21}^2 N_b},$$

where we use the fact that $-\max_\phi \ell_{\mathcal{D}}(\theta^t, \phi) \le -V(\theta^t)$. Telescoping the inequalities from $\tau = 0$ to $\tau = N$, we arrive at

$$\mathbb{E}[V(\theta^t) - \ell_{\mathcal{D}}(\theta^t, \phi^t(N))] \le \rho^N(V(\theta^t) - \ell_{\mathcal{D}}(\theta^t, \phi^t(0))) + \frac{1-\rho^N}{1-\rho}\left(\frac{L_{22}\sigma^2}{2L_{21}^2 N_b}\right).$$

PL-condition implies quadratic growth, we also know that $V(\theta^t) - \ell_{\mathcal{D}}(\theta^t, \phi^t(N)) \le \mathbb{E}_\xi \frac{1}{2\mu}\|\nabla_\phi \mathcal{L}_{\mathcal{D}}(\theta^t, \phi_\xi^t(N), \xi)\|^2 \le \frac{1}{2\mu}G^2$, by Assumption F.3,

$$\|\phi_\xi^*(\theta^t) - \phi_\xi^t(N)\|^2 \le \frac{2}{\mu}(\mathcal{L}_{\mathcal{A}}(\theta^t, \phi_\xi^*, \xi) - \mathcal{L}_{\mathcal{A}}(\theta^t, \phi_\xi^t(N), \xi))$$

$$\le \frac{2|c|}{\mu}\left|\mathcal{L}_{\mathcal{D}}(\theta^t, \phi_\xi^*, \xi) - \mathcal{L}_{\mathcal{D}}(\theta^t, \phi_\xi^t(N), \xi)\right|$$

Hence, with Jensen inequality and choice of $N_{\mathcal{A}}$ and $N_b$,

$$\mathbb{E}[\|z_t\|] = \mathbb{E}[\|\nabla_\theta V(\theta^t) - \mathbb{E}_\xi \nabla_\theta \mathcal{L}_{\mathcal{D}}(\theta^t, \phi_\xi^t(N_{\mathcal{A}}), \xi)\|]$$

$$\le L_{12}\mathbb{E}[\|\phi_\xi^t(N_{\mathcal{A}}) - \phi_\xi^*\|]$$

$$\le L_{12}\sqrt{\frac{2|c|}{\mu}\mathbb{E}[V(\theta^t) - \ell_{\mathcal{D}}(\theta^t, \phi^t(N_{\mathcal{A}}))]}$$

$$\le L_{12}\sqrt{\frac{|c|}{\mu^2}\rho^{N_{\mathcal{A}}}G^2 + (1-\rho^{N_{\mathcal{A}}})\frac{|c|L_{22}^2\sigma^2}{\mu L_{21}^2 N_b}}.$$

Now we adjust the size of $N_{\mathcal{A}}$ and $N_b$ to make $\mathbb{E}[\|z_t\|]$ small enough, to this end, we set

$$\rho^{N_{\mathcal{A}}}\frac{|c|G^2}{\mu^2} \le \frac{\varepsilon^4 L^2}{32 D_V^2 (2V_\infty + LD_\Theta)^4 \bar{L}}$$

$$\frac{|c|L_{22}^2\sigma^2}{L_{21}^2 N_b} \le \frac{\varepsilon^4 L^2 \mu^2}{32 D_V^2 (2V_\infty + LD_\Theta)^4 \bar{L}},$$

which further indicates that

$$N_{\mathcal{A}} \ge \frac{1}{\log \rho^{-1}} \log \frac{32 D_V^2 (2V_\infty + LD_\Theta)^4 \bar{L}|c|G^2}{L^2 \mu^2 \varepsilon^4}$$

$$N_b \ge \frac{32\mu L_{21}^2 D_V^2 (2V_\infty + LD_\Theta)^4}{|c|L_{22}^2\sigma^2 \bar{L}L\varepsilon^4}.$$

In the setting above, it is not hard to verify that

$$\mathbb{E}[\|z_t\|] \le \frac{L\varepsilon^2}{4D_V(2V_\infty + LD_\Theta)^2} \le \varepsilon.$$

Also note that $\|\nabla_\phi \mathcal{L}_{\mathcal{A}}(\theta^t, \phi_\xi^t(N_{\mathcal{A}}), \xi)\| = \|\nabla_\phi \mathcal{L}_{\mathcal{A}}(\theta^t, \phi_\xi^t(N_{\mathcal{A}}), \xi) - \nabla_\phi \mathcal{L}_{\mathcal{A}}(\theta^t, \phi_\xi^*, \xi)\|$, given the proper choice of $N_{\mathcal{A}}$ and $N_b$, one has

$$\mathbb{E}\|\nabla_\phi \mathcal{L}_{\mathcal{A}}(\theta^t, \phi_\xi^t(N_{\mathcal{A}}), \xi) - \nabla_\phi \mathcal{L}_{\mathcal{A}}(\theta^t, \phi_\xi^*, \xi)\|$$

$$\le L_{21}\mathbb{E}[\|\phi_\xi^t(N_{\mathcal{A}}) - \phi_\xi^*\|] \le \frac{L\varepsilon^2}{4D_V(2V_\infty + LD_\Theta)^2} \le \varepsilon,$$

which indicates the $\xi$-wise inner loop stability. $\qquad\square$

Now we are ready to provide the convergence guarantee of the first-order outer loop.

**Theorem F.8.** *Under Assumption F.6, Assumption F.3, and Assumption F.2, let the stepsizes be,* $\kappa_{\mathcal{A}} = \frac{1}{L_{22}}$, $\kappa_{\mathcal{D}} = \frac{1}{L}$, *if* $N_{\mathcal{D}}, N_{\mathcal{A}}$, *and* $N_b$ *are large enough,*

$$N_{\mathcal{D}} \geq N_{\mathcal{D}}(\varepsilon) \sim \mathcal{O}(\varepsilon^{-2}) \quad N_{\mathcal{A}} \geq N_{\mathcal{A}}(\varepsilon) \sim \mathcal{O}(\log \varepsilon^{-1}), \quad N_b \geq N_b(\varepsilon) \sim \mathcal{O}(\varepsilon^{-4})$$

*then there exists* $t \in \mathbb{N}$ *such that* $(\theta^t, \{\phi_\xi^t(N_{\mathcal{A}})\}_{\xi \in \Xi})$ *is* $\varepsilon$-*meta-FOSE.*

*Proof.* According to the update rule of the outer loop, (here we omit the projection analysis for simplicity)

$$\theta^{t+1} - \theta^t = \frac{1}{L}\hat{\nabla}_\theta \ell_{\mathcal{D}}(\theta^t, \phi^t(N_{\mathcal{A}})),$$

one has, due to unbiasedness assumption, let $(\mathcal{F}_t)_{0 \leq t \leq N_{\mathcal{D}}}$ be the filtration generated by $\sigma(\theta^t | k \leq t)$

$$\mathbb{E}[\langle \nabla_\theta \ell_{\mathcal{D}}(\theta^t, \phi^t(N_{\mathcal{A}})), \theta^{t+1} - \theta^t \rangle | \mathcal{F}_t] = \frac{1}{L}\mathbb{E}[\|\nabla_\theta \ell_{\mathcal{D}}(\theta^t, \phi^t(N_{\mathcal{A}}))\|^2 | \mathcal{F}_t]$$
$$= L\mathbb{E}\|\theta^{t+1} - \theta^t\|^2 | \mathcal{F}_t],$$

which leads to

$$\mathbb{E}[\langle \nabla_\theta \ell_{\mathcal{D}}(\theta^t, \phi^*), \theta^{t+1} - \theta^t \rangle | \mathcal{F}_t] = \mathbb{E}[\langle z_t, \theta^t - \theta^{t+1} \rangle | \mathcal{F}_t] + L\mathbb{E}[\|\theta^{t+1} - \theta^t\|^2\|].$$

Since $V(\cdot)$ is $L$-Lipschitz smooth,

$$\mathbb{E}[V(\theta^t) - V(\theta^{t+1})] \leq \mathbb{E}[\langle \nabla_\theta V(\theta^t), \theta^t - \theta^{t+1} \rangle] + \frac{L}{2}\mathbb{E}[\|\theta^{t+1} - \theta^t\|^2]$$

$$\leq \mathbb{E}[\langle z_t, \theta^{t+1} - \theta^t \rangle] - \mathbb{E}[\langle \nabla_\theta \ell_{\mathcal{D}}(\theta^t, \phi^t(N_{\mathcal{A}})), \theta^{t+1} - \theta^t \rangle] + \frac{L}{2}\mathbb{E}[\|\theta^{t+1} - \theta^t\|^2] \quad \text{(F12)}$$

$$\leq \mathbb{E}[\langle z_t, \theta^{t+1} - \theta^t \rangle] - \frac{L}{2}\mathbb{E}[\|\theta^{t+1} - \theta^t\|^2].$$

Fixing a $\theta \in \Theta$, let $e_t := \langle \nabla_\theta \ell_{\mathcal{D}}(\theta^t, \phi^t(N_{\mathcal{A}})), \theta - \theta^t \rangle$, we have

$$\mathbb{E}[e_t | \mathcal{F}_t] = L\mathbb{E}[\langle \theta^{t+1} - \theta^t, \theta - \theta^t \rangle | \mathcal{F}_t]$$
$$= \mathbb{E}[\langle \nabla_\theta \ell_{\mathcal{D}}(\theta^t, \phi^t(N_{\mathcal{A}})) - \nabla_\theta V(\theta^t), \theta^{t+1} - \theta^t \rangle + \langle \nabla_\theta V(\theta^t), \theta^{t+1} - \theta^t \rangle]$$
$$\quad + L\mathbb{E}[\langle \theta^{t+1} - \theta^t, \theta - \theta^{t+1} \rangle] \quad \text{(F13)}$$
$$\leq \mathbb{E}[(\|z_t\| + V_\infty + LD_\Theta)\|\theta^{t+1} - \theta^t\|]$$

By the choice of $N_b$, we have, since $V_\infty = \max\{\max_\theta \|\nabla V(\theta)\|, 1\}$,

$$\mathbb{E}[\|z_t\|] \leq L_{12}\mathbb{E}[\|\phi^N - \phi^*\|] \leq \frac{L\varepsilon^2}{4D_V(2V_\infty + LD_\Theta)} \leq V_\infty.$$

Thus, the relation equation F13 can be reduced to

$$\mathbb{E}[e_t] \leq (2V_\infty + LD_\Theta)\mathbb{E}[\|\theta^{t+1} - \theta^t\|].$$

Telescoping equation F12 yields

$$-D_V \leq \mathbb{E}[V(\theta^0) - V(\theta^{N_{\mathcal{D}}})] \leq D_\Theta \sum_{t=0}^{T-1}\mathbb{E}[\|z_t\|] - \frac{L}{2(2V_\infty + LD_\Theta)^2}\mathbb{E}[\sum_{t=0}^{T-1}\mathbb{E}[e_t^2 | \mathcal{F}_t].$$

Thus, setting $N_{\mathcal{D}} \geq \frac{4D_V(2V_\infty + LD_\Theta)^2}{L\varepsilon^2}$, and then by Lemma F.7, we obtain that,

$$\frac{1}{N_{\mathcal{D}}}\sum_{t=0}^{N_{\mathcal{D}}-1}\mathbb{E}[e_t^2] \leq \frac{\varepsilon^2}{2} + \frac{2D_V(2V_\infty + LD_\Theta)^2}{LN_{\mathcal{D}}} \leq \varepsilon^2$$

which implies there exists $t \in \{0, \ldots, N_{\mathcal{D}} - 1\}$ such that $\mathbb{E}[e_t^2] \leq \varepsilon^2$.

$\square$

## F.3 Generalization to Unseen Attacks

In the online adaptation phase, the pre-trained meta-defense may be exposed to attacks unseen in the pre-training phase, which poses an out-of-distribution (OOD) generalization issue to the proposed meta-SG framework. Yet, Proposition F.9 and Proposition F.13 assert that meta-SG is generalizable to the unseen attacks, given that the unseen is not distant from those seen. The formal statement is deferred to Appendix F, and the proof mainly targets those unseen non-adaptive attacks for simplicity.

**Proposition F.9** (OOD Generalization Informal Statement). *Consider sampled attack types* $\xi_1, \ldots, \xi_m$ *during the pre-training and the unseen attack type* $\xi_{m+1}$ *in the online stage. The generalization error is upper-bounded by the "discrepancy" between the unseen and the seen attacks* $C(\xi_{m+1}, \{\xi_i\}_{i=1}^m)$.

Our main goal is to quantify the value discrepancy under an attack type that is out of empirical distribution. We consider attack types $\xi_1, \ldots, \xi_m$ to be empirically sampled from distribution $Q(\cdot)$ during the pre-training stage, and an unseen attack type $\xi_{m+1}$ in the online stage. The quantification of distance $C(\xi_{m+1}, \{\xi_i\}_{i=1}^m)$ relies on the total variation,

**Definition F.10** (total variation). For two distributions $P$ and $Q$, defined over the sample space $\Omega$ and $\sigma$-field $\mathcal{F}$, the total variation between $P$ and $Q$ is $\|P - Q\|_{TV} := \sup_{U \in \mathcal{F}} |P(U) - Q(U)|$.

The celebrated result shows the following characterization of total variation,
$$\|P - Q\|_{TV} = \sup_{f: 0 \le f \le 1} \mathbb{E}_{x \sim P}[f(x)] - \mathbb{E}_{x \sim Q}[f(x)].$$

Let the fixed attack policies $\phi_i$, $i = 1, \ldots, m+1$ corresponding to each attack type. To formalize the generalization error, for each $\theta \in \Theta$, we define populational values
$$\hat{V}(\theta) := \frac{1}{m} \sum_{i=1}^m \mathbb{E}_{\tau \sim q_i^\theta} J_\mathcal{D}(\theta - \eta \hat{\nabla}_\theta J_\mathcal{D}(\tau), \phi_i, \xi_i)$$
$$\hat{V}_{m+1}(\theta) := \mathbb{E}_{\tau \sim q_{m+1}^\theta} J_\mathcal{D}(\theta - \eta \hat{\nabla}_\theta J_\mathcal{D}(\tau), \phi_{m+1}, \xi_{m+1})$$

where $q_i^\theta(\cdot) : (S \times A \times S)^{H-1} \times S \to [0, 1]$ is the trajectory distribution determined by state dependent policies $\pi_\mathcal{D}(\cdot|s; \theta)$, $\pi_\mathcal{A}(\cdot|s; \phi_i, \xi_i)$ and transition kernel $\mathcal{T}$. Since $q_i^\theta$ is factorizable, we have Lemma F.11 to eliminate $\|q_i^\theta - q_{m+1}^\theta\|_{TV}$ dependence on $\theta$ by upper bounding it using another pair of mariginal distributions.

**Lemma F.11.** *For any* $\theta \in \Theta$, *there exist marginals* $d_i, d_{m+1} : (S \times A_\mathcal{A} \times S)^{H-1} \times S \to [0, 1]$ *total variation* $\|q_i^\theta - q_{m+1}^\theta\|_{TV}$ *can be bounded by* $\|d_i - d_{m+1}\|_{TV}$.

*Proof.* By factorization, for a trajectory $\tau$, any $\theta \in \Theta$, and any type index $i = 1, \ldots, m+1$:
$$q_i^\theta(\tau) = \prod_{t=1}^{H-1} \pi_\mathcal{D}(a_\mathcal{D}^t|s_t; \theta) \prod_{t=1}^{H-1} \pi_\mathcal{A}(a_\mathcal{A}^t|s_t, \phi_i, \xi_i) \prod_{t=1}^{H-1} \mathcal{T}(s_{t+1}|s_t, a_t),$$
thus, by the inequality of product measure,
$$\|q_i^\theta - q_{m+1}^\theta\|_{TV} \le \sum_{t=1}^{H-1} \underbrace{\|\pi_\mathcal{D}(\cdot|s_t; \theta) - \pi_\mathcal{D}(\cdot|s_t; \theta)\|_{TV}}_{0} + \|d_i - d_{m+1}\|_{TV},$$
where $d_i$ and $d_{m+1}$ are the residue factors after removing $\pi_\mathcal{A}(\cdot|s_t; \theta)$. $\qquad\square$

**Assumption F.12.** *For any* $\xi \in \Xi$ *and* $\phi_\xi$, *the function* $J_\mathcal{D}(\theta, \phi_\xi, \xi)$ *is* $G$-*Lipschitz continuous w.r.t.* $\theta \in \Theta$;

**Proposition F.13.** *Under assumption 3.2 and certain regularity conditions, fixing a policy* $\theta \in \Theta$, *we have, there exist some marginal distribution of*
$$|\hat{V}_{m+1}(\theta) - \hat{V}(\theta)| \le C(d_{m+1}, \{d_i\}_{i=1}^m),$$
*where the constant* $C$ *depending on the total variation between* $d_{m+1}$ *and* $\{d_i\}_{i=1}^m$:
$$C(d_{m+1}, \{d_i\}_{i=1}^m) := \frac{2\eta G^2}{m} \sum_{i=1}^m \|d_{m+1} - d_i\|_{TV} + \frac{1 - \gamma^H}{1 - \gamma} \left\|d_{m+1} - \frac{1}{m} \sum_{i=1}^m d_i\right\|_{TV},$$
*here,* $G$ *is the Lipschitz parameter of* $J_\mathcal{D}$ *w.r.t. both* $\theta$.

*Proof.* We start with the decomposition of the generalization error, for an arbitrary attack type $\xi_i$, $i = 1, \ldots, m$, fixing a policy $\theta \in \Theta$ determines jointly with each $\phi_i$ the trajectory distribution $q_i^\theta$. Denoting the one-step adaptation policy $\theta'(\tau) = \theta - \eta \nabla J_\mathcal{D}(\tau)$ as a function of trajectory $\tau$, we have the following decomposition,

$$\hat{V}_{m+1}(\theta) - \hat{V}(\theta) = \mathbb{E}_{\tau_{m+1} \sim q_{m+1}^\theta} J_\mathcal{D}(\theta'(\tau_{m+1}), \phi_{m+1}, \xi_{m+1}) - \frac{1}{m} \sum_{i=1}^m \mathbb{E}_{\tau_i \sim q_i^\theta} J_\mathcal{D}(\theta'(\tau_i), \phi_i, \xi_i)$$

$$= \underbrace{\mathbb{E}_{\tau_{m+1} \sim q_{m+1}^\theta} J_\mathcal{D}(\theta'(\tau_{m+1}), \phi_{m+1}, \xi_{m+1}) - \frac{1}{m} \sum_{i=1}^m \mathbb{E}_{\tau_{m+1} \sim q_{m+1}^\theta} J_\mathcal{D}(\theta'(\tau_{m+1}), \phi_i, \xi_i)}_{(i)}$$

$$+ \underbrace{\frac{1}{m} \sum_{i=1}^m \mathbb{E}_{\tau_{m+1} \sim q_{m+1}^\theta} J_\mathcal{D}(\theta'(\tau_{m+1}), \phi_i, \xi_i) - \frac{1}{m} \sum_{i=1}^m \mathbb{E}_{\tau_i \sim q_i^\theta} J_\mathcal{D}(\theta'(\tau_i), \phi_i, \xi_i)}_{(ii)}.$$

We assume $(\tau_{m+1}, \tau_i)$ is drawn from a joint distribution which has marginals $q_{m+1}^\theta$ and $q_i^\theta$ and is corresponding to the maximal coupling of these two. Then,

$$\tau_{m+1} \sim q_{m+1}^\theta, \quad \tau_i \sim q_i^\theta, \quad \mathbb{P}(\tau_{m+1} \neq \tau_i) = \|q_i^\theta - q_{m+1}^\theta\|_{TV},$$

if $\tau_{m+1}$ disagrees with $\tau_i$, for $(ii)$, we have, since $J_\mathcal{D}^\theta$ is Lipschitz with respect to $\theta$,

$$\|J_\mathcal{D}(\theta'(\tau_{m+1}), \phi_i, \xi_i) - J_\mathcal{D}(\theta'(\tau_i), \phi_i, \xi_i)\|$$

$$\leq \eta G \|\hat{\nabla}_\theta J_\mathcal{D}(\tau_{m+1}) - \hat{\nabla}_\theta J_\mathcal{D}(\tau_i)\|$$

$$\leq 2\eta G^2,$$

as a result, denoting the maximal coupling of $q_{m+1}^\theta$ and $q_i^\theta$ as gives,

$$[\mathbb{E}_{\tau_{m+1} \sim q_{m+1}^\theta} J_\mathcal{D}(\theta'(\tau_{m+1}), \phi_i, \xi_i) - \mathbb{E}_{\tau_i \sim q_i^\theta} J_\mathcal{D}(\theta'(\tau_i), \phi, \xi_i)]$$

$$= \mathbb{E}_{(\tau_{m+1}, \tau_i) \sim \prod(q_{m+1}^\theta, q_i^\theta)} [J_\mathcal{D}(\theta'(\tau_{m+1}), \phi_i, \xi_i) - J_\mathcal{D}(\theta'(\tau_i), \phi, \xi_i)]$$

$$\leq 2\eta G^2 \|q_{m+1}^\theta - q_i^\theta\|_{TV} \leq 2\eta G^2 \|d_i - d_{m+1}\|_{TV},$$

where the last inequality is due to Lemma F.11. Averaging the $m$ empirical $\xi_i$'s yeilds the result:

$$(ii) \leq \frac{2\eta G^2}{m} \sum_{i=1}^m \|d_i - d_{m+1}\|_{TV}.$$

Since the trajectory distribution is a product measure, the difference between $q_i^\theta$ and $q_{m+1}^\theta$ only lies by attacker's type, $\|q_{m+1}^{\theta'(\tau_{m+1})} - q_i^{\theta'(\tau_{m+1})}\|_{TV} = \|q_{m+1}^\theta - q_i^\theta\|_{TV} \leq \|d_{m+1} - d_i\|_{TV}$.

Now we bound $(i)$, for ease of exposition we let $q'' = q_{m+1}^{\theta'(\tau_{m+1})}$ and $q_i' := q_i^{\theta'(\tau_{m+1})}$. By the finiteness of total trajectory reward $R(\tau)$ for any trajectory $\tau$, $R(\tau) \leq \frac{1-\gamma^H}{1-\gamma}$, hence,

$$(i) = \mathbb{E}_{\tau_{m+1} \sim q_{m+1}^\theta} J_\mathcal{D}(\theta'(\tau_{m+1}), \phi_{m+1}, \xi_{m+1}) - \frac{1}{m} \sum_{i=1}^m \mathbb{E}_{\tau_{m+1} \sim q_{m+1}^\theta} J_\mathcal{D}(\theta'(\tau_{m+1}), \phi_i, \xi_i)$$

$$= \mathbb{E}_{\tau_{m+1} \sim q_{m+1}^\theta} \left[ \mathbb{E}_{\tau'' \sim q''} R_\mathcal{D}(\tau'') - \frac{1}{m} \sum_{i=1}^m \mathbb{E}_{\tau_i' \sim q_i'} R_\mathcal{D}(\tau_i') \right]$$

$$\leq \mathbb{E}_{\tau_{m+1} \sim q_{m+1}^\theta} \frac{1-\gamma^H}{1-\gamma} \|q_{m+1}'' - \frac{1}{m} \sum_{i=1}^m q_i'\|_{TV}$$

$$\leq \frac{1-\gamma^H}{1-\gamma} \|d_{m+1} - \frac{1}{m} \sum_{i=1}^m d_i\|_{TV}.$$

$\square$

## G   A Game-theoretic Perspective on Meta Equilibrium

This section offers further justification for the meta-equilibrium in (meta-SE), and we argue that meta-equilibrium provides a data-driven approach to address incomplete information in dynamic games. Note that information asymmetry is prevalent in the adversarial machine learning context, where the attacker enjoys an information advantage (e.g., the attacker's type). The proposed meta-equilibrium notion can shed light on these related problems beyond the adversarial FL context.

We begin with the insufficiency of Bayesian Stackelberg equilibrium defined as the solution to the bilevel optimization in equation BSE in handling information asymmetry, a customary solution concept in security studies [35].

$$\max_{\theta \in \Theta} \mathbb{E}_{\xi \sim Q(\cdot)}[J_{\mathcal{D}}(\theta, \phi_\xi^*, \xi)] \quad \text{s.t. } \phi_\xi^* \in \arg\max J_{\mathcal{A}}(\theta, \phi, \xi), \forall \xi \in \Xi. \tag{BSE}$$

One can see from equation BSE that such an equilibrium is of ex-ante type: the defender's strategy is determined before the game starts. It targets a "representative" attacker (an average of all types). As the game unfolds, new information regarding the attacker's private type is revealed (e.g., through the global model updates). However, this ex-ante strategy does not enable the defender to adjust its strategy as the game proceeds. Using game theory language, the defender fails to handle the emerging information in the interim stage.

To create interim adaptability in this dynamic game of incomplete information, one can consider introducing the belief system to capture the defender's learning process on the hidden type. Let $I^t$ be the defender's observations up to time $t$, i.e., $I^t := (s^k, a_{\mathcal{D}}^k)_{k=1}^t s^{t+1}$. Denote by $\mathcal{B}$ the belief generation operator $b^{t+1}(\xi) = \mathcal{B}[I^t]$. With the Bayesian equilibrium framework, the belief generation can be defined recursively as below

$$b^{t+1}(\xi) = \mathcal{B}[s^t, a_{\mathcal{D}}^t, b^t] := \frac{b^t(\xi)\pi_{\mathcal{A}}(a_{\mathcal{A}}^t|s^t; \xi)\mathcal{T}(s^{t+1}|s^t, a_{\mathcal{A}}^t, a_{\mathcal{D}}^t)}{\sum_{\xi'} b^t(\xi')\pi_{\mathcal{A}}(a_{\mathcal{A}}^t|s^t; \xi')\mathcal{T}(s^{t+1}|s^t, a_{\mathcal{A}}^t, a_{\mathcal{D}}^t)}. \tag{G1}$$

Since $b^t$ is the defender's belief on the hidden type at time $t$, its belief-dependent Markovian strategy is defined as $\pi_{\mathcal{D}}(s^t, b^t)$. Therefore, the interim equilibrium, also called Perfect Bayesian Equilibrium (PBE) [17] is given by a tuple $(\pi_{\mathcal{D}}^*, \pi_{\mathcal{A}}^*, \{b^t\}_{t=1}^H)$ satisfying

$$\pi_{\mathcal{D}}^* = \arg\max \mathbb{E}_{\xi \sim Q} \mathbb{E}_{\pi_{\mathcal{D}}, \pi_{\mathcal{A}}^*}[\sum_{t=1}^H r_{\mathcal{D}}(s^t, a_{\mathcal{D}}^t, a_{\mathcal{A}}^t)b^t(\xi)]$$

$$\pi_{\mathcal{A}}^* = \arg\max \mathbb{E}_{\pi_{\mathcal{D}}, \pi_{\mathcal{A}}}[\sum_{t=1}^H r_{\mathcal{A}}(s^t, a_{\mathcal{D}}^t, a_{\mathcal{A}}^t)], \forall \xi, \tag{PBE}$$

$\{b^k\}_{k=1}^H$ satisfies $(G1)$ for realized actions and states.

In contrast with (BSE), this perfect Bayesian equilibrium notion (PBE) enables the defender to make good use of the information revealed by the attacker, and subsequently adjust its actions according to the revealed information through the belief generation. From a game-theoretic viewpoint, both (PBE) and (meta-SE) create strategic online adaptation: the defender can infer and adapt to the attacker's private type through the revealed information since different types aim at different objectives, hence, leading to different actions. Compared with PBE, the proposed meta-equilibrium notion is better suited for large-scale complex systems where players' decision variables can be high-dimensional and continuous, as argued in the ensuing paragraph.

To achieve the strategic adaptation, PBE relies on the Bayesian-posterior belief updates, which soon become intractable as the denominator in equation G1 involves integration over high-dimensional space and discretization inevitably leads to the curse of dimensionality. Despite the limited practicality, PBE is inherently difficult to solve, even in finite-dimensional cases. It is shown in [6] that the equilibrium computation in games with incomplete information is NP-hard, and how to solve for PBE in dynamic games remains an open problem. Even though there have been encouraging attempts at solving PBE in two-stage games [36], it is still challenging to address PBE computation in generic Markov games.

