# OpenReview forum: "Meta Stackelberg Game: Robust Federated Learning against Adaptive and Mixed Poisoning Attacks"
_NeurIPS.cc/2024/Conference — Submitted to NeurIPS 2024_

### Official Review · Reviewer_s9YV · 2024-07-04

**Soundness:** 3
**Presentation:** 3
**Contribution:** 3
**Rating:** 5
**Confidence:** 3

**Summary:**

The paper titled "Meta Stackelberg Game: Robust Federated Learning against Adaptive and Mixed Poisoning Attacks" proposes a new approach to enhancing the security of Federated Learning (FL) systems. The paper identifies that existing FL defenses are inadequate against adaptive and mixed attacks. To address this, they introduce a Meta Stackelberg Game (meta-SG) framework, which employs a Bayesian Stackelberg Markov game (BSMG) and a meta-learning approach. This framework aims to provide a robust and adaptive defense mechanism against various poisoning attacks, including model poisoning and backdoor attacks. The proposed method is theoretically proven to converge to an equilibrium efficiently and is empirically validated to perform well against strong adversarial attacks.

**Strengths:**

+ Introducing the Meta Stackelberg Game (meta-SG) framework is innovative, offering a new perspective on defending against adaptive and mixed attacks in FL.

+ The paper provides a solid theoretical foundation, proving that the proposed algorithm converges to a first-order ε-equilibrium, proving the method's efficiency.

+ Extensive experiments demonstrate the effectiveness of the meta-SG framework, showing significant improvements in defense against various attack types compared to existing methods.

+ The meta-learning component allows the defense mechanism to adapt dynamically to different attack scenarios, enhancing its robustness in uncertain environments.

**Weaknesses:**

- The proposed approach seems computationally intensive, requiring significant resources for pre-training and adaptation, which might limit its practicality in real-world applications. Although it proves that it can converge in Theorem 3.3, it would be beneficial to have empirical evidence, such as the method's run-time overhead.

- While the framework is tested against several attack types, the scope of attacks considered might not cover all possible real-world adversarial strategies, limiting the generalizability of the results. The paper especially makes it unclear what attacks are used in pre-training, whether they are the same, and how different they are compared to the real FL environment when testing and generating results.

- The proposed method's scalability to larger and more diverse FL environments remains unclear, especially given its computational demands.

**Questions:**

Given the above points, I have some questions that need to be addressed:

1. How does the meta-SG framework scale with increasing clients and more complex models? Have you tested its performance in larger FL setups?

2. How does the framework handle adaptive attack types not seen during the pre-training phase? Are there any limits to the adaptive attacks it can defend against?

3. Have you considered applying the meta-SG framework to domains other than image classification, such as natural language processing or time-series analysis in FL?

---

> ### Author Rebuttal · Authors · 2024-08-07
>
> 1. computationally intensive
>
> We stress that our proposed method deals with mixed attacks of unknown and uncertain types, which is beyond the scope of other baselines that focus on specific attacks. Since our problem setup is more complicated, it is not surprising that more computational resources are required.
>
> From a theoretical viewpoint, our meta-Stackelberg learning (meta-SL) in the pre-training phase amounts to a two-time scale fully first-order policy gradient algorithm for solving a bi-level stochastic programming problem. In terms of sample complexity, the meta-SL has reached the SOTA efficiency in bi-level algorithms. This computation complexity is not because of our design fallacy but due to the bi-level structure of the meta-Stackelberg equilibrium defense, which we consider as the best fit to combat the defender's incomplete information on the mixed attack types involving strong adaptive attacks.
>
> 2. generalization
>
> We agree with the reviewer that the considered attacks do not cover all possibilities. However, as presented in Section 3.1, the pre-training considers white-box RL attacks as surrogates for those unseen strong attacks in the real world. These white-box RL attacks present the worst-case scenario in the training, and the resulting defense can generalize to other attacks. We report the generalization experiment in Table 8 Appendix D, where the pre-trained defense achieves satisfying performance against unseen adaptive attacks. Please also refer to Table 3 in Appendix which showcase the set of attacks and defenses employed during pre-training and online adaptation and their related figures/tables.
>
> 4. scalability
>
> All the baseline defenses conducted similar or smaller scale experiments as we did. We further tested larger-scale experiment with meta-SG against LMP on CIFAR-10 including 1000 clients where 100 clients been selected each round (online adaptation only). The test is conducted on AWS p2.16xlarge instance with 16 GPUs, 64 vCPUs, 732 Mem (GiB) and 192 GPU Memory (GiB). We implement parallel computing on 10 client simultaneously. The average global model accuracy after 500 rounds reach 0.6954 which close to our result in Table 1.
>
> 5. unseen adaptive attack type
>
> As shown in line 87, page 2, and also in "Online Adaptation and Execution'' in line 799, page 19, the defender collects trajectories of model updates to fine-tune its pre-trained meta policy using gradient adaptation, no matter whether the attack is seen or unseen in the pre-training (the defender has no access to the attack type in the online stage). As certified in Proposition 3.4 (and also Proposition F.13), the meta-SG defense performance degradation when facing unseen attacks is upper-bounded by the "distance'' between those seen and the unseen. The "distance'' is defined by the total variation between trajectories produced by the attacks. In other words, given an acceptable defense performance drop, we can quantify the maximum "distance'' of an unseen attack to the set of seen, which becomes the limit of the meta-SG defense. To elaborate on this generalization property empirically, we test our framework with unsee attacks, and the results are in Table 8 on page 22, which presents satisfying defense performance.
>
> 6. meta-SG for other application
>
> Please refer to our discussion in the paper's conclusion section. Our defense framework is general and can potentially be applied to other domains beyond image classification. We encourage researchers with different expertise to join in applying this framework or its modified version to more domains.

---

### Official Review · Reviewer_4AkQ · 2024-07-13

**Soundness:** 3
**Presentation:** 4
**Contribution:** 3
**Rating:** 7
**Confidence:** 3

**Summary:**

The paper presents a game theoretic model for robust federated learning. The technique is composed of pre-training and online adaptation. During pre-training, a meta-policy for the defender is solved as a Bayesian Stackelberg Markov game. The defense policy is further polished during the online adaptation stage.

**Strengths:**

The Stackelberg game is designed to counter unknown/uncertain attacks by an adaptive adversary.

Theoretical bounds for sample complexity are provided.

Empirical results demonstrate the effectiveness of the proposed technique.

**Weaknesses:**

The main weakness is the slight violation of privacy as the technique needs a portion of ground truth data from the clients. This has been disused as the limitations in the paper.

Minor comments:

On page 2, "including mixed attacks ," ---> extra space

**Questions:**

What is the statistical significance of the results shown in Table 1? Without the standard error, it is unclear whether the proposed technique is indeed superior to other existing techniques. I understand there is no room in the table, at least it should be mentioned in the text.

**Limitations:**

Privacy violation is mentioned as a limitation of the technique. It's unclear whether future development can remove the dependence on the client-side ground truth.

---

> ### Author Rebuttal · Authors · 2024-08-07
>
> Please refer to rebuttal to Reviewer a9ia for privacy concern. We have address the minor comments. We currently didn't calculate the statistical significance (need more computation resources). In most experiments, we fix initial model and all random seeds for fair comparisons. Client-side defense need thoroughly design which leave to our future work.

---

> ### Author Response · Authors · 2024-08-08
>
> I apologize for the brevity of our initial response, which was due to time constraints. Thank you for your valuable feedback. We have carefully addressed the minor comments you provided.
>
> - Regarding the privacy concerns, please refer to our response to Reviewer a9ia's comments.
> - We acknowledge the importance of statistical significance and have made efforts to address this in our study. Specifically, we conducted 20 trials for the experiments shown in Figures 11 (c) and (d) to generate the error bars. However, due to limitations in computational resources, we were unable to calculate statistical significance for all of our results. We plan to address this in future work when we have access to additional resources. For the other graphs and tables, we fixed the initial model and all random seeds across experiments to ensure fair comparisons.
> - Additionally, we recognize that client-side defenses are crucial for enhancing privacy and security. We agree that this is an area that requires thorough design and development, and it is indeed part of our future research agenda.

---

> > ### Comment · Reviewer_4AkQ · 2024-08-11
> > **Response to authors' rebuttal**
> >
> > Thanks for the explanations. I'll keep my score.

---

### Official Review · Reviewer_Ui9w · 2024-07-13

**Soundness:** 2
**Presentation:** 2
**Contribution:** 2
**Rating:** 4
**Confidence:** 4

**Summary:**

The authors propose a defense mechanism in federated learning that has adaptability inspired from meta learning. The authors formulates a Bayesian Stackelberg Markov game (BSMG), focusing on addressing poisoning attacks of unknown or uncertain types. The authors propose an equilbrium inspired by meta learning and then look at a local version of that. Empirical evaluations are done on MNIST and CIFAR.

**Strengths:**

This paper seems to have interesting results.

**Weaknesses:**

The writing is sloppy in many places and quite a few things are unexplored.
Federated learning's primary motivation is privacy, so the privacy loss from a core small dataset must be analyzed. The authors do acknowledge the loss but IMO that is not enough.

Behind all the motivation of federated learning, the core idea is in the equilibrium proposed - IMO, exploring this equilibrium in more detail would make the paper stronger (in fact, the problem makes sense even in simpler adversarial learning problems).

Is Def 3.1 just a differential equilibrium, in the style of "Implicit Learning Dynamics in Stackelberg Games: Equilibria Characterization, Convergence Analysis, and Empirical Study" but missing second order conditions?
Why are there no second order conditions? Just first order may not induce a local equilibrium, which is a meaningful equilibrium to achieve. This is where even more meaning needs to be discovered for the first meta-SE. The authors compare it to PBE in the appendix, but the notations of belief consistency and sequential rationality is what makes PBE (SE) convincing. Without any such (or similar) notion, a new equilibrium in a dynamic setting is not principled.

I do not understand why the ball $B(\theta^*)$ is used with a bound of 1? What is special about 1? (same for the other ball)
Proposition written informally (and no explanation) in the main paper does not make sense (e.g., Prop 3.4).

There are many typos when I started to look in appendix:
1) Eq F6, $\tilde{l}$ should have two inputs
2) Line 959, the parameters of $\tilde{l}$ is lost, without this the equations with $\theta'$ on left and no $\theta'$ on right is not well-defined.
3) I do not understand how the equation in line 964 came about - first it is said that it is an inequality but what is written is an equality.

Overall, typos do not inspire confidence.
Also, any defense mechanism should itself be subject to attack with the adversary possessing knowledge of defense mechanism - I do not see that in experiments.

**Questions:**

Please respond to review

---

> ### Author Rebuttal · Authors · 2024-08-07
>
> 1. privacy concern
>
> Please refer to rebuttal to Reviewer a9ia
>
> 2. meta-Stackelberg equilibrium
>
> Indeed, the core of the proposed meta-Stackelberg framework lies in the meta-Stackelberg equilibrium (meta-SE). The essence of meta-SE is to create the strategic adaptation in the interim stage (online) in a data-driven approach (online gradient updates) when the defender has incomplete information about the attacker. Such a data-driven approach deviates from the conventional Bayesian approach in PBE.  Since the defender's and attacker's objective functions in FL are non-convex, the exact analysis of the meta-SE is challenging due to the bi-level structure of the equilibrium definition and the dynamic nature of FL processes (i.e., a Markov game). We agree that a simpler adversarial learning problem (e.g., supervised learning), which can be modeled as a static Bayesian game, may lead to a thorough theoretical analysis. However, such a direction is a digression from our FL problem. Even though we are unable to provide strong theoretical characterizations of the proposed meta-SE, we conduct empirical studies to demonstrate the advantage of meta-SE over conventional Bayesian equilibrium (BSE defined in line 251). As shown in Figure 1 in the author rebuttal pdf,  the BSE policy  (red curve) does not display effective adaptation within the 500 rounds, whereas the meta-SE policy (blue curve) quickly adapts to the LMP attack.
>
> 3. Is Def 3.1 just a differential equilibrium?
>
> Def 3.1 is not a differential equilibrium. The differential equilibrium (Def 4) in the reference is a sufficient condition, whereas our definition presents a first-order necessary condition. These first-order conditions, along with the positive-definiteness of the Hessian matrix, construct the optimality conditions for a local solution for the meta-SE, which may not exist even in the zero-sum cases [1, proposition 6]. Hence, we consider the necessary conditions to guarantee the existence.
>
> We agree with the reviewer that belief consistency and sequential rationality provide a solid theoretical foundation for PBE when players are Bayesian: they maintain a prior distribution over the uncertainty (type in our case) and make decisions based on the posterior. However, as we argue in Appendix G, computing Bayesian posterior is intractable in large-scale FL systems with neural network models. Our meta-SE essentially discards the Bayesian approach in handling incomplete information games. Instead, we resort to the online gradient adaptation to process the information feedback without computing the posterior. Even though meta-SE is not as well-grounded as PBE in game theory, it does provide a non-Bayesian data-driven alternative in handling incomplete information that is suitable for complex multi-agent systems such as FL.
>
> 4. the ball $\mathcal{B}$
>
> We consider the unit ball to avoid introducing more notation; in fact, we can use any ball with radius $r$. In the setting where the $r$-radius ball $\|\theta-\theta^*\|\leq r$ falls within the space $\Theta$ (i.e., unconstrained setting), the condition simply implies $\|\nabla\_\theta \mathcal{L}\_\mathcal{D} (\theta^*, \phi^*_\xi, \xi) \| \leq \varepsilon/r$. To see this, we let $\theta = \theta^* +r \frac{\nabla\_{\theta} \mathcal{L}\_\mathcal{D} (\theta^*, \phi^*_\xi, \xi)}{\|\nabla\_{\theta} \mathcal{L}\_\mathcal{D} (\theta^*, \phi^*\_\xi, \xi)\|}$, and then direct calculation gives the upper-bound on the gradient norm. We simply let $r=1$ in this work, which is a common practice in optimization literature [2].
>
> Regarding Proposition 3.4, the explanation is that the generalization error, which is the difference of expected returns under a learned defense policy with different attack types, can be controlled by the total variations between trajectory distributions induced by the sampled attack types and unseen attack types. The ``discrepancy'' concerns additional notations that might make it less readable. Appendix F.3 gives more detailed explanations.
>
> 5. typos in Appendix
>
> (1) $\tilde{\ell}\_{\mathcal{D}}$ is a concave function augmented from $ \ell\_{\mathcal{D}}$ that takes three arguments: $\theta, \theta^{\prime}, \text{ and } \phi^{\prime}$, and there was a typo in $\ell\_{\mathcal{D}}(\theta, \phi)$, which should be $\ell_{\mathcal{D}}(\theta, \phi^{\prime})$, which likely caused the confusion.
>
> (2) It should be $\tilde{\ell}\_{\mathcal{D}}(\theta; (\theta^{\prime}, \phi^{\prime}) ) $, thanks to the reviewer multiple typos have been corrected.
>
> (3) Apology for this typo. "inequality" should be "equalities". We basically augmented the functions here to make them satisfy the fixed-point theorem, which guarantees the existence. We then leveraged the equalities to show that the equilibrium of game $ (\tilde{\ell}\_{\mathcal{D}}, \tilde{\ell}\_{\xi})$ is also the equilibrium of game $( \ell\_{\mathcal{D}},  \ell\_{\xi})$.
>
>
>
> [1] Chi Jin, Praneeth Netrapalli, and Michael I. Jordan. What is Local Optimality in Nonconvex-Nonconcave Minimax Optimization? ArXiv:1902.00618, 2019.
>
> [2] Nouiehed, Maher, Maziar Sanjabi, Tianjian Huang, Jason D. Lee, and Meisam Razaviyayn. "Solving a class of non-convex min-max games using iterative first order methods." Advances in Neural Information Processing Systems 32 (2019).

---

> > ### Comment · Reviewer_Ui9w · 2024-08-11
> >
> > I am not an expert on federate learning, so I leave the significance of that to other reviewers. But, I know game theory very well and I am not satisfied (and curious) about this new sort of equilibrium idea proposed without really exploring what it means in the sequential setting.
> >
> > The privacy angle still is a question - I feel used in prior work precedent is not enough.

---

> ### Author Response · Authors · 2024-08-12
> **Thanks for the reviewer's comment**
>
> We appreciate Reviewer Ui9w's insightful suggestions. We would like to clarify that sequential rationality does not apply to the meta equilibrium due to the data-driven gradient adaptation. Our proposed meta-equilibrium concept is a mixed-strategy equilibrium instead of a behavioral-strategy equilibrium if we represent the Markov game in an extensive form. The defense policy $\pi_D(a^t|s^t;\theta)$, even though takes in $s^t$ at each time step, is not a behavioral strategy because $s^t$ does not correspond to an information set in this incomplete information dynamic game. In the Bayesian framework (e.g., PBE), the information set given by the history is equivalent to a Bayesian posterior belief. A behavioral strategy must take in the information sets (or beliefs) to determine the actions. Since computing Bayesian posteriors is intractable in large FL systems, our framework discards the Bayesian approach and embarks on a data-driven method to handle interim information. The learned policy $\pi_D(a^t|s^t;\theta)$ only determines a probability measure over the action sequence $\{a^1,a^2,\ldots, a^H\}$ to be played. In summary, the proposed meta-equilibrium is not a subgame perfect equilibrium in the Markov game because one cannot define the perfection condition (sequential rationality) [Def. 8.2, 1] without defining the information set in the Bayesian framework. Our motivation is to trade sequential rationality for computational viability and efficiency.
>
>
> We believe that it is practical for the server to collect a small, clean training dataset, known as the root dataset, for the learning task. For instance, Google uses its employees to type with Gboard, creating a root dataset for its federated next-word prediction [2]. This root dataset may or may not align with the distribution of the overall training data used for the learning task.
>
>
>
> [1] Fudenberg, D. and Tirole, J. Game Theory. MIT Press, 1991.
>
> [2] Federated Learning: Collaborative Machine Learning without Centralized Training Data. [Online]. Available: https://ai.googleblog.com/2017/04/federated-learning-collaborative.htm

---

### Official Review · Reviewer_a9ia · 2024-07-14

**Soundness:** 2
**Presentation:** 2
**Contribution:** 2
**Rating:** 4
**Confidence:** 1

**Summary:**

This paper considers the problem of backdoor/poisoning attacks in federated learning (FL). In this setting, a single attacker has control over all malicious clients trying to employ different attack types (on each controlled client). This paper aims to create a defense mechanism against such adaptive attackers. To this end, the paper proposes a game-theoretic approach, which contains two stages: (1) Pre-training stage: before the FL environment, the defender first learns a good defense policy by simulating that environment using a small amount of truthful data against a simulated attacker with known possible actions (e.g. attack types used), and (2) Online-adaptation stage: the defender leverages the pre-trained policy and adapt it against the attacker at the real FL environment. This paper demonstrates the effectiveness of their proposed mechanism, as well as considers ablation studies where the previous assumptions do not meet: adaptation to attack methods at real FL environment are different (but similar) from ones seen at the pre-training stage.

**Strengths:**

The paper is well-written and easy to follow.

**Weaknesses:**

I have some concerns, mostly about the practicality of the assumptions:

1. Assumption on the accessibility of data: the paper assumes that in the pre-training phase, the defender has access to a small amount of data, which is used to model the data distribution of clients using generative models. (1) It goes against privacy. (2) It is not clear that a small amount of data is representative enough to model client data distribution. (3) Things will be even more complicated if each client has its sub-population (data) that is (reasonably) different from each other. In this case, which malicious clients the attacker has control over will matter.

2. Assumption of the similarity of attack types at the online adaptation stage: the paper assumes that the attack types in a real FL environment, though unseen, should be similar to those of the pre-training phase. This seems impractical, especially in a white-box setting, where the attacker will try to leverage the property of the defense mechanism to create an adaptive attack that is specified for that defense mechanism (see Carlini's works).

I also have concerns about the experiments:
1. Datasets and models used: Would it be possible to use more practical datasets and models instead of MNIST/CIFAR10 and ResNet-18? I would like to see results when the data distribution is complex enough that generative models cannot easily learn with a small amount of data. Right now, the amount of data used is still considerable, considering the dataset used. This would make the assumption of the accessibility to a small amount of data in the pre-training phase more persuasive.

Some comments on writing/paper organizations.
1. In Table 1, please highlight which results are the best. It would be more readable and easier for comparison.
2. In Figure 2, it might be better to show smoothed curves.
3. In Appendix F, before each theorem/lemma/assumption, it would be better if an intuition/proof sketch for each one is provided. Also, if the proof technique/assumption is standard, please mention the corresponding references.
4. In the Conclusion, I think it would be more honest if explicitly stated that the major limitation of this paper lies in the practicability of the assumption. Right now, I only see privacy concerns mentioned, which is misleading.

(Not really a weakness) It could be nice if source code is included (might be using an anonymous repo).

Overall, I think this is a good paper if ignoring the practicability of the assumptions on the attack types/data (on the accessibility in the pre-training phase/distribution of accessed data in the pre-training phase/distribution of data in each client). I also did not find an experiment where the attacker leverages the information about defense mechanisms to instantiate a better attack scheme (white box attack). It is known that many proposed defense mechanisms failed in this scenario, though it seems obvious that it will go against the similarity assumption on the attack types at the online adaptation phase. However, I am also aware of the hardness of defense tasks in adversarial machine learning and it is good to have some initial results even under strong assumptions. I will try my best to be reasonable. Maths were not checked carefully, I will try to go into the details in the rebuttal phase.

**Questions:**

See Weaknesses above.

**Limitations:**

See Weaknesses above.

---

> ### Author Rebuttal · Authors · 2024-08-07
>
> 1. practicality of the assumptions
>
> (1) In the conclusion of the paper, we have discussed the potential privacy issue of our approach, pointed out our initial efforts to mitigate this concern, and outlined a future direction to address it in a more principled way (i.e., client-side defense). We also note that some recent studies on developing robust defenses in FL [1][2] also rely on a small amount of root data. (2) Please refer to the empirical results in Appendix D (``Importance of inverting/reversing methods'') and the theoretical generalization result in [3] that characterizes the impact of inaccurate data distribution on the performance of RL-based attacks, which directly applies to our RL defense in the face of a fixed attack. (3) Please refer to our non-i.i.d. experiment results in Appendix D (Figure 11 (d)).
>
> 2. Assumption of the similarity of attack types at the online adaptation stage
>
> We first clarify that the similarity between two types is measured by the closeness between trajectories produced by different attacks rather than their attack methods and configurations. Due to the page limit,  We postpone the presentation to Appendix F.3.  For example, the white-box attack proposed by the reviewer and the RL adaptive attack in the domain, even though may employ quite different attack algorithms, are considered similar if they generate similar trajectories in the sense of total variation since they both aim to best respond to the defense mechanism.
>
> As presented in Section 3.1, we do consider white-box RL attacks in the pre-training stage as a surrogate for strong attacks (e.g., the one suggested by the reviewer). These white-box RL attacks present the worst-case scenario the defender can encounter, and the resulting defense can be robust to other weaker attacks.  This idea of preparing for the worst naturally leads to the Stackelberg game model proposed in our paper.  We test the meta-SG defense against several unseen white-box adaptive attacks, and the results are in Table 8 Appendix D. The key observation is that our framework delivers satisfying performance.
>
> 3. complex datasets/model
>
> MNIST, CIFAR-10, and ResNet-18 are commonly used in federated learning literature. Currently, we lack the computational resources to experiment with larger datasets and more complex model structures, which we plan to address in future work.
>
> 4. writing/paper organizations
>
> We have address (1)(3) and discussed (4) above. For point (2), we fix the initial model and use fixed random seeds for fair comparisons in most experiments and we include error bars in Figure 11 to account for variability.
>
>
> [1] Xiaoyu Cao, Minghong Fang, Jia Liu, and Neil Zhenqiang Gong. Fltrust: Byzantine-robust federated learning via trust bootstrapping. In Network and Distributed System Security (NDSS) Symposium, 2021.
>
> [2] Yinbin Miao, Ziteng Liu, Hongwei Li, Kim-Kwang Raymond Choo, and Robert H Deng. Privacy-preserving byzantine-robust federated learning via blockchain systems. IEEE Transactions on Information Forensics and Security, 17:2848–2861, 2022.
>
> [3] Henger Li, Xiaolin Sun, and Zizhan Zheng. Learning to attack federated learning: A model based reinforcement learning attack framework. In Advances in Neural Information Processing Systems, 2022.

---

> ### Comment · Reviewer_a9ia · 2024-08-08
> **Reply to the rebuttal**
>
> - **On the dataset used in this work**: my concern is not that the paper lacks of experiments with large datasets, but:
>    -  In other federated learning works, it might be fine to use small datasets like MNIST and CIFAR10. However, there is a critical component in this paper, which is modeling data distribution of the clients given limited access to data. How can we know that if the data distribution is more complex, then we still can effectively (and sufficiently so that it does not affect the performance of the frameworks) the data distribution with limited data? This is my concern.
>
>    - Moreover, I am also aware couple of works that use other (a bit more) complex datasets in federated learning (Tiny ImageNet), for example [1]. It would be nice if the experiments for those datasets were incorporated into this work.
>
> Overall, I find the rebuttal not convincing. Though I decided to keep my original rating, I would reduce my confidence score to 1.
>
> References
>
> [1] Dung Thuy Nguyen et al. IBA: Towards Inversible Backdoor Attacks in Federated Learning. NeurIPS'23.

---

> ### Author Response · Authors · 2024-08-12
> **Modeling data distribution in complex dataset**
>
> Thank you for the practical advice.
>
> We would like to first clarify that in addition to generative modeling in pre-training, we also utilize the inverting gradient (IG) method [1] in the online FL stage to infer the clients' data distribution (Appendix C lines 801-804), aiming to bridge the gap in data distribution. Due to space limitations, we moved the discussion of IG to Appendix C lines 783-791. A recent paper [2] shows that IG can successfully reconstruct images from ImageNet-1K from gradient data. While implementing more powerful generative models and advanced inverting gradient methods is beyond the scope of this work, our meta-SG framework can be integrated with them to handle more complex datasets.
>
> We further note that our approach can still work even if the learned distribution deviates slightly from the true distribution. This was observed for RL-based attacks in [Thm 1, 3], but a similar result also holds for RL-based defenses.
>
> [1] Geiping, J., Bauermeister, H., Dröge, H., & Moeller, M. (2020). Inverting gradients-how easy is it to break privacy in federated learning? Advances in neural information processing systems.
>
> [2] Hatamizadeh, Ali, et al. "Gradvit: Gradient inversion of vision transformers." Proceedings of the IEEE/CVF Conference on Computer Vision and Pattern Recognition. 2022.
>
> [3] Henger Li, Xiaolin Sun, and Zizhan Zheng. Learning to attack federated learning: A model based reinforcement learning attack framework. In Advances in Neural Information Processing Systems, 2022.

---

### Official Review · Reviewer_HbAX · 2024-07-19

**Soundness:** 3
**Presentation:** 2
**Contribution:** 3
**Rating:** 5
**Confidence:** 3

**Summary:**

The paper addresses the vulnerabilities of Federated Learning (FL) systems to various adversarial attacks, including model poisoning and backdoor attacks. The proposed solution is a Meta Stackelberg Game (meta-SG) framework designed to offer robust and adaptive defenses against these attacks. The approach formulates adversarial federated learning as a Bayesian Stackelberg Markov game (BSMG) and employs a meta-learning method to find optimal defense strategies. Theoretical analyses show the algorithm's efficiency in convergence, and empirical results demonstrate its effectiveness against different attack types.

**Strengths:**

1. The proposed framework considers both untargeted and targeted attacks.

2. The paper uses RL to realize online adaptation which is close to the real world.

3. Inspired by meta-learning, the method is robust to unknown attacks.

4. In the pre-training phase, the paper uses generated data to decrease the concern of privacy leakage.

5. The paper provides sufficient experimental results.

**Weaknesses:**

1. Could you explain more about the necessity of adding the gradient adaptation (from BSE to meta-SE)? Although BSE is ex-ante when knowing the distribution $Q$, the model $\theta$ is changed during training and it could capture emerging information. Could you provide empirical results comparing BSE and meta-SE to show the advantage of meta-SE?

2. Considering adaptive/mixed attacks, the paper misses two relevant frameworks: MixTailor [1] and RobustTailor [2]. They can adjust aggregation methods during training. Especially, RobustTailor also simulates a game between the defender and the attacker, and it proposes a mixed attack. This kind of method could be included in experiments as a baseline.

3. Because the whole method is complicated with 2 stages, comparing computational cost with other baselines is necessary.




[1] Ramezani-Kebrya, Ali, Iman Tabrizian, Fartash Faghri, and Petar Popovski. "Mixtailor: Mixed gradient aggregation for robust learning against tailored attacks." arXiv preprint arXiv:2207.07941, 2022.

[2] Xie, Wanyun, Pethick, Thomas, Ramezani-Kebrya, Ali, and Cevher, Volkan. "Mixed Nash for Robust Federated Learning." Transactions on Machine Learning Research. 2023.

**Questions:**

1. Are both white-box and black-box settings used in the pre-training stage? Adding the proposed methods explicitly in Figure 1 might be more clear.

**Limitations:**

The authors mentioned limitations in Section 5. The main one is the privacy concern.

---

> ### Author Rebuttal · Authors · 2024-08-07
>
> 1. meta-SE vs BSE
>
> We compare the BSE policy $\theta_{BSE}$ and the meta-SE $\theta_{meta}$ from an information feedback viewpoint. The BSE policy uses the current global model $s^t=w^t_g$ to determine the defense action: $\pi_\mathcal{D}(a^t_\mathcal{D}|s^t, \theta_{BSE})$. This policy is Markovian and only uses the emerging information of the current global model to output actions targeting an average attack. In contrast, meta-SE uses online trajectory $\tau$ (a sequence of local/global model updates) to first generate policy adaptation $\theta\_{adapted}=\theta\_{meta}+\eta \hat{\nabla}\_\theta J\_\mathcal{D}(\tau)$.
> Then, the adapted policy uses the current global model to determine the defense action: $\pi_\mathcal{D}(a^t_\mathcal{D}|s^t, \theta_{adapted})$. Naturally, $\tau$ incorporated into $\theta_{adapted}$ reveals more information about the actual attack than $s^t$; hence, $\theta_{adapted}$ captures more emerging information than $\theta_{BSE}$ and is more tailored to the actual attack.
>
> We conduct additional experiments comparing the defense performance under BSE and meta-SE; see Figure 1 in the author rebuttal pdf. The pre-training follows the same setup as in the paper (see Figure 2 and associated discussions in the paper). The key message is that the BSE policy (red curve) does not display effective adaptation within the 500 rounds, whereas the meta-SE policy (blue curve) quickly adapts to the LMP attack.
>
> 2. two relevant frameworks
>
> We believe these papers, at least in their current form, are not suitable to be included as baselines in our experiments due to the following reasons. 1) The two papers address dynamically switching between existing defenses, which is not particularly useful for addressing our mixed attack problem, where multiple types of attacks (e.g., untargeted model poisoning and backdoor attacks) can occur simultaneously in a single round of FL. The focal point of our work is not about **choosing** a defense but rather how to **combine** them effectively. 2) The major contribution of our meta-SG framework is to use meta learning to address the defender's incomplete information of mixed attacks (similar to [1][2]) and adaptively coordinate a set of defenses (beyond [1][2]).
>
> 3. computational cost
>
> Our meta-Stackelberg framework deals with the mixed attacks of unknown and uncertain types, which is beyond the scope of other baselines focusing on specific attacks. Since our problem setup is more complicated, it is not surprising that more computational resources are required. We report the running time of pre-training and online implementation in Table 1 in the author rebuttal pdf. We stress that the execution time of the learned meta-Stackelberg is of the same level as the baselines.
>
> 4. white-box and black-box settings
>
> The pre-training stage only considers a white-box setting. In a simulated environment, we deal with a set of known attacks/environment parameters, collected from domain knowledge or historical experience. However, in the online stage, we must adapt to unknown attacks/environment parameters, which may be either previously encountered or entirely new. Please also refer to Table 3 in the Appendix to see the set of attacks and defenses employed during pre-training and online adaptation and their related figures/tables.
> Both settings are considered in our experiments: the main paper covers only the white-box setting as our default, while the black-box settings and corresponding experiments are detailed in Appendices C and D (see Figures 10 and 11).
>
> [1] Ramezani-Kebrya, Ali, Iman Tabrizian, Fartash Faghri, and Petar Popovski. "Mixtailor: Mixed gradient aggregation for robust learning against tailored attacks." arXiv preprint arXiv:2207.07941, 2022.
>
> [2] Xie, Wanyun, Pethick, Thomas, Ramezani-Kebrya, Ali, and Cevher, Volkan. "Mixed Nash for Robust Federated Learning." Transactions on Machine Learning Research. 2023.

---

> > ### Comment · Reviewer_HbAX · 2024-08-09
> > **Thanks for the rebuttal**
> >
> > Thanks for the authors' reply and explanation.
> >
> > I have the following question. Can backdoor-defending methods like NeuroClip be incorporated in other aggregation rules like FedAvg or Trimed Mean? If yes, why MixTailor or RobustTailor cannot be used?
> >
> > Currently, Meta-RL with extra pre-training only compared with some basic single aggregators. It'll be more convincing if it can be compared with some more 'smarter' aggregator.

---

> ### Author Response · Authors · 2024-08-12
> **Thanks for the follow-up question**
>
> Thanks for your valuable time in reviewing and constructive comments. The key to our framework's ability to integrate NeuroClip and Pruning is its provision of a scalable and efficient method (i.e., meta-RL) for tuning the hyperparameters of these defense mechanisms. We have implemented a 'smarter' aggregator, where we manually tune hyperparameters (see Table 7) for defenses and choose the optimal ones to be applied for ClipMed and FLTrust + NC in Table 1. When there are multiple hyperparameters to tune, it becomes impractical. For instance, with norm bounding, Trimmed Mean, and NeuroClip, the search space expands to the range of the norm bound threshold multiplied by the Trimmed percentage and the clipping range. This search space is continuous and grows exponentially as the number of hyperparameters increases. We naively tested MixTailor (and could not find an open-sourse implementation for RobustTailor) to dynamically transition from the existing defenses listed in Table 1. However, this approach yielded worse performance compared to using only FLtrsut + NC.
>
> Below we further clarify why the original versions of Mixtailor and RobustTailor are not suitable for comparison in our context due to the different considerations in problem setup, online adaptation, and scalability.
>
> $\textbf{Problem setup:}$ Both Mixtailor and RobustTailor consider defenses against a single white-box attack that is tailored to the honest training gradients. That is, the attack objective is to drag the aggregated gradient away from the honest one as much as possible. To counter such attacks, the two works explore the idea of randomization over aggregation rules, creating information asymmetry on the defense side.
>
> However, our work focuses on defending against a mixture of unknown/uncertain attacks. These attacks may not be tailored to the honest gradients, such as targeted attacks. Our framework uses game-theoretic meta-learning to address information asymmetry on the attack side, requiring the defense to be tailored online to unknown/uncertain attacks.
>
> $\textbf{Online Adaptation:}$ Due to different problem setups, MixTailor and RobustTailor do not consider and are unable to incorporate online adaptation. MixTailor simply picks an aggregation randomly, while RobustTailor approximately solves a minimax problem to get the sampling distribution over the set of aggregation rules. One can view the resulting defense as a worst-case defense without considering the actual attack methods. RobustTailor only uses the current gradient information at each round to determine the aggregation.
>
> In contrast, our proposed method considers online adaptation, which utilizes trajectories of online model updates (gradients) to implicitly identify the attack type (different attacks induce different trajectories). A trajectory contains more informative feedback than gradients at each round. Then, online gradient adaptation is derived using the trajectories.
>
> $\textbf{Scalability:}$ Both Mixtailor and RobustTailor considered randomization over a finite set of aggregation rules. Our experiments consider defense methods (e.g., aggregation rules) parameterized by continuous parameters, which are optimized by the proposed meta-RL algorithm using policy gradients, which can efficiently handle continuous action space. In contrast, to apply the two randomization methods, we need to first discretize the parameter space. The number of the discretized parameters grows exponentially with respect to the dimension (i.e., the number of hyperparameters for all defenses combined).

---

> > ### Comment · Reviewer_HbAX · 2024-08-13
> > **Thanks for the detailed response**
> >
> > Thanks for the detailed response, and it addressed my concerns about the other two frameworks.
> >
> > I'll keep my score because I'm not sure whether a little privacy leakage is acceptable in federated learning.

---

### Author Rebuttal · Authors · 2024-08-07

We extend our heartfelt gratitude to the reviewers for their invaluable questions, insightful comments, and constructive suggestions. We look forward to your inspiring thoughts. While the detailed responses are attached to reviewers' comments, we summarize some key updates and revisions here.

We thank reviewer HbAX for pointing us to the helpful references, which will be discussed in the related works section. We briefly mention that the two works may not be qualified for baselines since they address dynamic switching between defenses, whereas our method focuses on the adaptive combination of defenses. Additionally, as suggested by the reviewer, we report in Table 1 in the attached pdf the actual running time of our meta-SG and compare its execution time with other baselines. We also compare BSE and meta-SE empirically in Figure 1 to highlight the advantage of gradient adaptation.

We thank reviewer a9ia for suggestions on the paper writing. We have added relevant references, remarks, and proof
intuition for theoretical results in Appendix F. Meanwhile, we would like to point out that the adaptive white-box attack mentioned by the reviewer is considered in our work, and associated experimental results are in Table 8 Appendix D.

We thank reviewer Ui9w for the close inspection of our theoretical analysis. We have carefully checked Appendix F, corrected typos, and added additional remarks on important claims, results, and assumptions. Even though the FL setup considered in this work forbids a thorough theoretical characterization of meta-SE, we empirically compare the meta-SG framework with the conventional Bayesian Stackelberg game (BSG) model ( the equilibrium is defined in line 251). As presented in Figure 1 in the attached pdf, our meta-SG displays greater adaptability than BSG defense.

We thank reviewer s9YV's question on the scalability of meta-SG. We have tested larger-scale experiment with meta-SG against LMP on CIFAR-10 including 1000 clients where 100 clients been selected each round (online adaptation only).

---

> ### Author Response · Authors · 2024-08-13
> **Discussion about public dataset in FL**
>
> Dear Reviewers and AC,
>
> We would like to provide a more detailed discussion to address the privacy concerns related to accessing public datasets. In Federated Learning (FL), it is a common practice to use a small globally shared dataset to enhance robustness (see Section 3.1.1 of [1]). This dataset could come from a publicly available proxy source, a separate non-sensitive dataset, or a processed version of the raw data as suggested by [2]. The use of such public datasets is widely accepted in the FL community [1, 3, 4, 5]. For example, systems like Sageflow [6], Zeno [7], and Zeno++ [8] leverage public data at the server to address adversarial threats. Additionally, having public data available on the server supports collaborative model training with formal differential or hybrid differential privacy guarantees [1, 9, 10]. [10] introduces hybrid differential privacy, where some users voluntarily share their data. Many companies, such as Mozilla and Google, utilize testers with high mutual trust who opt into less stringent privacy models compared to the average end-user. It is also worth mentioning that one of the defenses that Reviewer HbAX mentioned, i.e., RobustTailor, also assumes a public dataset (see Remark 1 in the paper).
>
> [1] Kairouz, Peter, et al. "Advances and open problems in federated learning." Foundations and trends® in machine learning 14.1–2 (2021): 1-210.
>
> [2] Wang, Tongzhou, et al. "Dataset distillation." arXiv preprint arXiv:1811.10959 (2018).
>
> [3] Minghong Fang, Xiaoyu Cao, Jinyuan Jia, and Neil Gong. Local model poisoning attacks to Byzantine-robust federated learning. In USENIX Security Symposium, 2020.
>
> [4] Wenke Huang, Mang Ye, and Bo Du. Learn from others and be yourself in heterogeneous federated learning. In Conference on Computer Vision and Pattern Recognition (CVPR), 2022.
>
> [5] Naoya Yoshida, Takayuki Nishio, Masahiro Morikura, Koji Yamamoto, and Ryo Yonetani. Hybrid-FL for wireless networks: Cooperative learning mechanism using non-IID data. In IEEE International Conference on Communications (ICC), 2020.
>
> [6] Jungwuk Park, Dong-Jun Han, Minseok Choi, and Jaekyun Moon. Sageflow: Robust federated learning
> against both stragglers and adversaries. Advances in Neural Information Processing Systems (NeurIPS),
> 34:840–851, 2021.
>
> [7] Cong Xie, Sanmi Koyejo, and Indranil Gupta. Zeno: Distributed stochastic gradient descent with suspicion-based fault-tolerance. In International Conference on Machine Learning (ICML), 2019b.
>
> [8] Cong Xie, Sanmi Koyejo, and Indranil Gupta. Zeno++: Robust fully asynchronous SGD. In International Conference on Machine Learning (ICML), 2020b.
>
> [9] Xin Gu, Gautam Kamath, and Zhiwei Steven Wu. Choosing public datasets for private machine learning via gradient subspace distance. arXiv preprint arXiv:2303.01256, 2023.
>
> [10] Brendan Avent, Aleksandra Korolova, David Zeber, Torgeir Hovden, and Benjamin Livshits. BLENDER: Enabling local search with a hybrid differential privacy model. In USENIX Security Symposium, 2017.

---

### Decision · Program_Chairs · 2024-09-25

**Decision:**

Reject

**Comment:**

The authors have studied robustness of federated learning w.r.t. important and diverse types of attacks such as  model poisoning and backdoor attacks. The authors have formulated their problem as a Bayesian Stackelberg Markov game and applied a meta-learning method to obtain optimal defense strategies.

The reviews are mixed for this paper. The paper was thoroughly discussed,  and the reviewers have pointed to some major concerns. For example, Reviewer HbAX noticed that authors have missed comparison with two relevant frameworks. Reviewer Ui9w is concerned about the proposed equilibrium and the proofs. Reviewer a9ia is not convinced whether a complex data distribution can be efficiently modeled with a limited number of samples. Reviewer s9YV concerns on applicability of the proposed method in real-world settings given computational aspects. In sum, I think a careful revision of the paper to address reviewers’ concerns requires substantial work.